# An Information-Theoretic Evaluation of Generative Models in Learning Multi-modal Distributions

**Mohammad Jalali**
Department of Electrical and Computer Engineering
Isfahan University of Technology
mjalali@ec.iut.ac.ir

**Cheuk Ting Li**
Department of Information Engineering
The Chinese University of Hong Kong
ctli@ie.cuhk.edu.hk

**Farzan Farnia**
Department of Computer Science and Engineering
The Chinese University of Hong Kong
farnia@cse.cuhk.edu.hk

## Abstract

The evaluation of generative models has received significant attention in the machine learning community. When applied to a multi-modal distribution which is common among image datasets, an intuitive evaluation criterion is the number of modes captured by the generative model. While several scores have been proposed to evaluate the quality and diversity of a model's generated data, the correspondence between existing scores and the number of modes in the distribution is unclear. In this work, we propose an information-theoretic diversity evaluation method for multi-modal underlying distributions. We utilize the *Rényi Kernel Entropy (RKE)* as an evaluation score based on quantum information theory to measure the number of modes in generated samples. To interpret the proposed evaluation method, we show that the RKE score can output the number of modes of a mixture of sub-Gaussian components. We also prove estimation error bounds for estimating the RKE score from limited data, suggesting a fast convergence of the empirical RKE score to the score for the underlying data distribution. Utilizing the RKE score, we conduct an extensive evaluation of state-of-the-art generative models over standard image datasets. The numerical results indicate that while the recent algorithms for training generative models manage to improve the mode-based diversity over the earlier architectures, they remain incapable of capturing the full diversity of real data. Our empirical results provide a ranking of widely-used generative models based on the RKE score of their generated samples[1].

## 1 Introduction

Deep generative models trained by generative adversarial networks (GANs) [1] and diffusion models [2] have achieved impressive results in various unsupervised learning settings [3, 4, 5, 6]. Due to their success in generating image samples with high visual quality, the analysis of large-scale generative models has received great attention in the machine learning community. In particular, the evaluation of generative models has been extensively studied in the recent literature to understand the benefits and drawbacks of existing approaches to training generative models.

To address the evaluation task for generative models, multiple assessment scores have been proposed in the literature. The existing evaluation metrics can be divided into two general categories: 1)

---

[1]The code repository is available at https://github.com/mjalali/renyi-kernel-entropy.

37th Conference on Neural Information Processing Systems (NeurIPS 2023).

distance-based metrics including Fréchet Inception Distance (FID) [7] and Kernel Inception distance (KID) [8] scores, which measure a distance between the learned generative model and data distribution, 2) quality-based metrics including the Inception score [9], precision and recall scores [10, 11], and density and coverage scores [12], aiming to measure the quality and diversity of the generated samples based on the confidence and variety of labels assigned by a pre-trained neural net on ImageNet.

On the other hand, a different measure of diversity that is popular and standard in the information theory literature is entropy. The primary challenge with an entropy-based approach to the assessment of generative models is the statistical costs of entropy estimation in the typical high-dimensional spaces of image data, requiring a sample size exponentially growing with the dimension of data. Consequently, without further assumptions on the data distribution, it will be statistically and computationally infeasible to estimate the entropy value for high-dimensional image data.

In this work, we propose a novel information-theoretic approach for the diversity evaluation of generative models. Our proposed approach targets multi-modal data distributions comprised of several distinct modes, which is an applicable assumption to image datasets with a cluster-based structure due to their latent color and shape-based features. In this approach, we follow the entropy calculation in quantum information theory [13, 14] and utilize matrix-based Rényi entropy scores to evaluate the variety of samples produced by a generative model.

Considering the Rényi entropy in the Gaussian kernel space, we propose *Rényi kernel entropy (RKE)* and *relative Rényi kernel entropy (RRKE)* scores to measure the absolute and relative diversity of a multi-modal distribution with respect to the actual data distribution. We develop computationally efficient methods for estimating these entropy scores from empirical data with statistical convergence guarantees. We also prove that the RKE score will converge to standard differential entropy as the Gaussian kernel bandwidth approaches zero.

We provide an interpretation of the RKE score by deriving its closed-form expression for benchmark Gaussian mixture models (GMMs). In the GMM case, we show that the proposed RKE score reveals the number of Gaussian components in the underlying GMM, which motivates the method's application to general mixture distributions. We further extend this interpretation to mixture models with sub-Gaussian modes, e.g. modes with a bounded support set, to support the RKE score in more general settings. We discuss the numerical performance of the proposed score in synthetic mixture settings. Our numerical results demonstrate the fast convergence of the RKE score from a limited number of synthetic mixture data.

Next, we present the results of our numerical experiments on the evaluation of various state-of-the-art GAN and diffusion models using the RKE and RRKE metrics. Our numerical evaluation of the RKE score shows the lower diversity obtained by standard GAN models compared to real training data, which provides an information-theoretic numerical proof complementary to the birthday-paradox empirical proof for the same result in [15]. Furthermore, our empirical results suggest that the recent GAN and diffusion model architectures improve the mode-based variety of generated data over earlier GAN formulations. We can summarize the main contributions of this work as follows:

- Proposing an information theoretic approach to evaluate the diversity of generative models
- Developing computationally efficient methods to compute Rényi kernel entropy scores
- Providing theoretical and numerical support for the proposed evaluation methodology in the benchmark setting of Gaussian and sub-Gaussian mixture models
- Diversity evaluation of standard generative models using the information-theoretic method

## 2   Related Work

The evaluation of GAN-based generative models has been studied by a large body of related works. As surveyed in [16], several evaluation methods have been developed in the literature. The Inception score (IS) [9] uses the output of a pre-trained Inception-net model as features and proposes a score summing up the entropy-based diversity of assigned labels' distribution and confidence score averaged over the conditional labels' distribution. The modified IS (m-IS) in [17] substitutes the KL-divergence term in IS with a cross entropy term, which helps m-IS capture diversity within images from a certain class. Unlike these works, our proposed approach bases on matrix-based entropy scores which capture the number of clusters in a multi-modal distribution.

Also, several distance-based evaluation metrics have been proposed in the deep learning literature. The Wasserstein Critic [18] attempts to approximate the Wasserstein distance between the real and generated samples. The Fréchet Inception Distance (FID) [7] measures a distance based on the embedding of the last layer of the pre-trained Inception-net, where it fits multivariate Gaussian models to real and generated data and calculates their Fréchet distance [19]. [20] conduct a comprehensive comparison of several standard GAN architectures based on IS and FID scores, discussing the similarities and differences of the GANs' performance. As another variant of FID, [21] suggest a bias-free estimation of FID using quasi-Monte Carlo integration. In another related work, [8] propose Kernel Inception Distance (KID) as the squared maximum mean discrepancy (MMD) between two distributions. Adversarial accuracy and divergence scores in [22] utilize two classifiers to compute the closeness of distributions of real and fake data conditioned on category labels from the two classifiers. Unlike the above metrics, our proposed relative Rényi entropy score focuses on the number of common modes between real and fake data, and hence reduces the statistical complexity of estimating the entropy.

The diversity vs. quality tradeoff of GANs' generated samples has also been studied in multiple related works. [15] examine the diversity of GANs' data through a birthday paradox-based approach and suggest that the support set size of GANs' data could be smaller than real training data. The precision and recall evaluation by [10] assigns a two-dimensional score where precision is defined as the portion of fake data that can be generated by real distribution while recall is defined as the portion of real data that can be generated by the generative model. The improved precision and recall in [11] further address the sensitivity disadvantages of these scores by estimating the density function via the k-nearest neighbour method. Also, the density and coverage scores [12] provide a more robust version of precision and recall metrics to outliers. Additionally, [23] proposed a 3-dimensional metric, that measures the fidelity, diversity and generalization of models. We note that our work offers a complementary approach to the diversity evaluation for GANs, and since it directly estimates the matrix-based entropy from data, it requires a comparatively smaller sample size for proper estimation.

## 3 Preliminaries

### 3.1 Kernel-based Feature Maps and Representation

Throughout the paper, we use $\mathbf{X} \in \mathcal{X}$ to denote the data vector. Also, we denote the kernel feature map by $\phi : \mathbb{R}^t \to \mathbb{R}^d$ which gives us the kernel function $k : \mathbb{R}^t \times \mathbb{R}^t \to \mathbb{R}$ as the inner product of the corresponding feature vectors: $k(\mathbf{x}, \mathbf{y}) = \langle \phi(\mathbf{x}), \phi(\mathbf{y}) \rangle$. Given $n$ training data $\mathbf{x}_1, \ldots, \mathbf{x}_n$ we use $\Phi$ to denote the normalized kernel feature map-based data matrix, i.e.,

$$\Phi = \frac{1}{\sqrt{n}} \begin{bmatrix} \phi(\mathbf{x}_1) \\ \vdots \\ \phi(\mathbf{x}_n) \end{bmatrix}.$$

Then, the kernel matrix $K \in \mathbb{R}^{n \times n}$ whose $(i,j)$th entry will be $\frac{1}{n} k(\mathbf{x}_i, \mathbf{x}_j)$ will be identical to $K = \Phi \Phi^\top$. Observe that, by definition, $K$ will be a positive semi-definite (PSD) matrix, possessing positive eigenvalues $\lambda_1, \ldots, \lambda_n$ where the non-zero eigenvalues are shared with the empirical kernel covariance matrix $C = \Phi^\top \Phi$. We call a kernel function normalized if $k(\mathbf{x}, \mathbf{x}) = 1$ for every $\mathbf{x} \in \mathcal{X}$. A standard example of a normalized kernel function is the Gaussian kernel with bandwidth parameter $\sigma$ defined as:

$$k_\sigma(\mathbf{x}, \mathbf{y}) := \exp\Big(-\frac{\|\mathbf{x} - \mathbf{y}\|_2^2}{2\sigma^2}\Big).$$

For every normalized kernel function, the eigenvalues of kernel matrix $K$ (and also empirical kernel covariance $C$) will be non-negative and add up to 1 as the trace of $K$ will be 1. As a result, $K$'s eigenvalues can be interpreted as a probability sequence.

### 3.2 Rényi Entropy for PSD Matrices

A standard extension of the entropy concept to PSD matrices is the matrix-based Rényi entropy [14]. The Rényi entropy of order $\alpha > 0$ for a PSD matrix $A \in \mathbb{R}^{d \times d}$ with eigenvalues $\lambda_1, \ldots, \lambda_d$ is defined as

$$\mathrm{RE}_\alpha(A) := \frac{1}{1 - \alpha} \log\big(\mathrm{Tr}(A^\alpha)\big) = \frac{1}{1 - \alpha} \log\Big(\sum_{i=1}^d \lambda_i^\alpha\Big),$$

where Tr denotes the trace operator. A commonly-used special case which we use throughout the paper is the Rényi entropy of order $\alpha = 2$ defined as $\text{RE}_2(A) = \log\big(1/\sum_{i=1}^{d}\lambda_i^2\big)$.

**Proposition 1.** *For every PSD $A \in \mathbb{R}^d$, the following holds where $\|\cdot\|_F$ denotes the Frobenius norm:*

$$\text{RE}_2(A) = \log\big(1/\text{Tr}(AA^\top)\big) = \log\big(1/\|A\|_F^2\big).$$

### 3.3 Relative Rényi Entropy

To measure the relative diversity of a matrix $A \in \mathbb{R}^{d \times d}$ with respect to another matrix $B \in \mathbb{R}^{d \times d}$, one can use the sandwiched relative Rényi entropy of order $\alpha$ [24] defined as

$$\text{RRE}_\alpha(A, B) = \frac{1}{\alpha - 1} \log\Big( \text{Tr} \Big( \big(B^{\frac{1-\alpha}{2\alpha}} A B^{\frac{1-\alpha}{2\alpha}}\big)^\alpha \Big) \Big).$$

A widely-used special case is the relative entropy of order $\alpha = \frac{1}{2}$ which is commonly called the Fidelity score in quantum information theory. The definition of the Fidelity score is

$$\text{RRE}_{1/2}(A, B) := -2\log\Big( \text{Tr}\big(\sqrt{B^{1/2} A B^{1/2}}\big)\Big).$$

We note that the relative Rényi entropy of order $\alpha = 2$ requires an invertible matrix $B$ which may not hold in the applications to multi-modal distributions with rank-deficient kernel covariance matrices as discussed in the next sections.

## 4 A Diversity Metric for Multi-modal Distributions

### 4.1 Kernel-based Rényi Entropy Scores

Given the kernel matrix $K$ computed using the data $\mathbf{x}_1, \ldots, \mathbf{x}_n$ of random vector $\mathbf{X}$, the empirical *Rényi kernel entropy (RKE)* of the observed data can be defined as

$$\widehat{\text{RKE}}_\alpha(\mathbf{X}) := \text{RE}_\alpha(K),$$

which, as discussed in [14], is a diversity measure of the data. This diversity measurement approach will be interpreted and justified in the next subsection. Note that if the dimension $d$ of the feature space is finite, since $K = \Phi\Phi^\top$ and empirical covariance matrix $\widehat{C} = \Phi^\top\Phi$ share the same eigenvalues, we have

$$\widehat{\text{RKE}}_\alpha(\mathbf{X}) = \text{RE}_\alpha(\widehat{C}) = \text{RE}_\alpha\bigg( \frac{1}{n} \sum_{i=1}^{n} \phi(\mathbf{x}_i)\phi(\mathbf{x}_i)^\top \bigg).$$

Therefore, we can see that the empirical Rényi kernel entropy $\widehat{\text{RKE}}_\alpha(\mathbf{X})$ is an estimate of the following quantity about the underlying distribution $P_X$ where $\mathbf{X}$ is sampled from, which we call the *Rényi kernel entropy* of $P_X$:

$$\text{RKE}_\alpha(\mathbf{X}) := \text{RE}_\alpha(C_X).$$

Here $C_X$ denotes the kernel covariance matrix of distribution $P_X$ defined as

$$C_X := \mathbb{E}_{P_X}\big[\phi(X)\phi(X)^\top\big] = \int P_X(x)\phi(x)\phi(x)^\top \mathrm{d}x.$$

Similarly, we can use the kernel-based relative Rényi entropy as a measure of joint diversity between distributions $P_X, P_Y$ of random vectors $\mathbf{X}, \mathbf{Y}$. Here, for random vectors $\mathbf{X}, \mathbf{Y}$ distributed according to $P_X, P_Y$, we define the *relative Rényi kernel entropy (RRKE$_\alpha$)* score as the order-$\alpha$ relative kernel entropy between their kernel covariance matrices $C_X, C_Y$:

$$\text{RRKE}_\alpha(\mathbf{X}, \mathbf{Y}) = \text{RRE}_\alpha(C_X, C_Y).$$

In order to estimate the above relative entropy score, we can use the empirical RRKE score between the empirical kernel covariance matrices for samples $\mathbf{x}_1, \ldots, \mathbf{x}_n$ from $P_X$ and samples $\mathbf{y}_1, \ldots, \mathbf{y}_m$ from $P_Y$:

$$\widehat{\text{RRKE}}_\alpha(\mathbf{X}, \mathbf{Y}) := \text{RRE}_\alpha\bigg( \frac{1}{n}\sum_{i=1}^{n}\phi(\mathbf{x}_i)\phi(\mathbf{x}_i)^\top \,, \, \frac{1}{m}\sum_{j=1}^{m}\phi(\mathbf{y}_j)\phi(\mathbf{y}_j)^\top \bigg).$$

In the rest of this section, we show how the kernel-based Rényi entropy score evaluated under a Gaussian kernel can relate to the number of modes of multi-modal distributions with sub-Gaussian components, and subsequently how the relative Rényi entropy score counts the number of joint modes between two multi-modal distributions with sub-Gaussian modes.

## 4.2 RKE as a Measure of the Number of Modes

We consider a multi-modal underlying distribution $P_X$ consisting of $k$ distinct modes. Here, our goal is to show that the order-2 RKE score under the Gaussian kernel can count the number of present modes in the distribution. To this end, we analyze the order-2 Rényi kernel entropy score of the kernel covariance matrix $C_X$ of mixtures of Gaussian and sub-Gaussian components and theoretically show that the RKE score reduces to the number of well-separated modes.

First, we derive the closed-form expression of the order-2 Rényi kernel entropy of a Gaussian mixture model under the Gaussian kernel. Here, we use the following notation to denote a $k$-component Gaussian mixture model where the $i$th component has frequency $\omega_i$, mean vector $\boldsymbol{\mu}_i$ and Covariance matrix $\Sigma_i$: $P_{\mathrm{GMM}(\boldsymbol{\omega},\boldsymbol{\mu},\Sigma)} = \sum_{i=1}^{k} \omega_i \mathcal{N}(\boldsymbol{\mu}_i, \Sigma_i)$. Also, for a positive definite matrix $A \in \mathbb{R}^{d \times d}$, we use $\|\cdot\|_A$ to denote the $A$-norm defined as $\|\mathbf{x}\|_A = \sqrt{\mathbf{x}^\top A \mathbf{x}}$.

**Theorem 1.** *Suppose that the distribution of $\mathbf{X}$ is given by the Gaussian mixture model $P_{\mathrm{GMM}(\boldsymbol{\omega},\boldsymbol{\mu},\Sigma)}$. Then, order-2 Rényi kernel score under the Gaussian kernel with bandwidth $\sigma$, denoted by $G_\sigma$, is*

$$\mathrm{RKE}_2^{G_\sigma}(\mathbf{X}) = -\log\bigg(\sum_{i=1}^{k}\sum_{j=1}^{k}\bigg[\omega_i\omega_j e^{-\frac{\|\boldsymbol{\mu}_i-\boldsymbol{\mu}_j\|_{A_{i,j}}^2}{\sigma^2}} \det\Big(I + \frac{2}{\sigma^2}(\Sigma_i + \Sigma_j)\Big)^{-\frac{1}{2}}\bigg]\bigg),$$

*where $A_{i,j}$ is defined as follows given the $d \times d$ identity matrix $I$:*

$$A_{i,j} := I - \big(I + 2\sigma^2\Sigma_i^{-1} - (I + 2\sigma^2\Sigma_j^{-1})^{-1}\big)^{-1} - \big(I + 2\sigma^2\Sigma_j^{-1} - (I + 2\sigma^2\Sigma_i^{-1})^{-1}\big)^{-1}$$
$$+ \big(2\sigma^2\Sigma_i^{-1} + 2\sigma^2\Sigma_j^{-1} + 4\sigma^4\Sigma_i^{-1}\Sigma_j^{-1}\big)^{-1} + \big(2\sigma^2\Sigma_i^{-1} + 2\sigma^2\Sigma_j^{-1} + 4\sigma^4\Sigma_j^{-1}\Sigma_i^{-1}\big)^{-1}.$$

*Proof.* We defer the proof to the Appendix. $\qquad\qquad\qquad\qquad\qquad\qquad\qquad\qquad\qquad\qquad$ $\square$

**Corollary 1.** *Suppose $\mathbf{X} \sim \mathrm{GMM}(\boldsymbol{\omega}, \boldsymbol{\mu}, \Sigma)$ follows from a Gaussian mixture model with isotropic covariance matrices $\Sigma_i = \sigma_i^2 I$. Defining the coefficients $a_{i,j} := 1 + \frac{2(\sigma_i^2+\sigma_j^2)}{\sigma^2}$, the RKE score will be*

$$\mathrm{RKE}_2^{G_\sigma}(\mathbf{X}) = -\log\bigg(\sum_{i=1}^{k}\sum_{j=1}^{k}\omega_i\omega_j a_{i,j}^{-\frac{d}{2}} e^{-\frac{a_{i,j}\|\boldsymbol{\mu}_i-\boldsymbol{\mu}_j\|_2^2}{\sigma^2}}\bigg).$$

Theorem 1 and Corollary 1 show that if $\sigma \gg \max_i \|\Sigma_i\|_{\mathrm{sp}}$, i.e. the kernel bandwidth dominates the spectral norm (maximum eigenvalue) of the component-wise covariance matrices, then $A_{i,j} \approx I$ and $a_{i,j} \approx 1$ and the RKE score will approximately be

$$\mathrm{RKE}_2^{G_\sigma}(\mathbf{X}) \approx -\log\Big(\sum_{i=1}^{k}\sum_{j=1}^{k}\omega_i\omega_j e^{-\frac{\|\boldsymbol{\mu}_i-\boldsymbol{\mu}_j\|_2^2}{\sigma^2}}\Big).$$

The next theorem shows the above approximation generalizes to mixtures of sub-Gaussian components, e.g. modes with bounded support sets, and also provides an approximation error bound.

**Theorem 2.** *Suppose that $\mathbf{X} \in \mathbb{R}^d$ has a mixture distribution $P_{\mathrm{SGMM}} = \sum_{i=1}^{k} \omega_i P_i$ where the $k$th component occurs with probability $\omega_k$ and has a sub-Gaussian distribution with parameter $\boldsymbol{\sigma}_i$, i.e. its moment-generating function (MGF) $M_{P_i}$ satisfies the following for every vector $\boldsymbol{\beta} \in \mathbb{R}^d$ given the mean vector $\boldsymbol{\mu}_i$ for $P_i$: $\mathbb{E}_{P_i}\big[\exp(\boldsymbol{\beta}^\top(\mathbf{X} - \boldsymbol{\mu}_i))\big] \leq \exp(\|\boldsymbol{\beta}\|_2^2\sigma_i^2/2)$. Then, the following approximation error bound holds where we define $\alpha_i^2 := 1 + 2\sigma_i^2/\sigma^2$:*

$$\bigg| \exp\Big(-\mathrm{RKE}_2^{G_\sigma}(\mathbf{X})\Big) - \sum_{i=1}^{k}\sum_{j=1}^{k}\omega_i\omega_j e^{-\frac{\|\boldsymbol{\mu}_i-\boldsymbol{\mu}_j\|_2^2}{\sigma^2}} \bigg| \leq \sqrt{\sum_{i=1}^{k} 8\omega_i\big(1 - \alpha_i^{-d}\big)}.$$

*Proof.* We defer the proof to the Appendix. $\qquad\qquad\qquad\qquad\qquad\qquad\qquad\qquad\qquad\qquad$ $\square$

More generally, we show that the RKE under a Gaussian kernel with a small bandwidth provides an estimate of the differential Rényi entropy, which connects the approach with a smoothed differential entropy estimator in high-dimensional settings.

**Theorem 3.** *Suppose that the underlying distribution of $X$ has a continuous probability density function $P_X$. The order-2 Rényi kernel score under Gaussian kernel with bandwidth $\sigma$ for this underlying distribution satisfies*

$$\lim_{\sigma \to 0} \left( \text{RKE}_2^{\text{G}_\sigma}(\mathbf{X}) + \frac{d}{2} \log(\pi \sigma^2) \right) = -\log \left( \int P_X(\mathbf{x})^2 \mathrm{d}\mathbf{x} \right),$$

*where the right hand side is the order-2 Rényi differential entropy of $P_X$.*

*Proof.* We defer the proof to the Appendix. □

### 4.3 RRKE as the Number of Common Modes

In order to measure the joint mode-based diversity, we propose the order-$\frac{1}{2}$ relative Rényi entropy score. We note that our choice of order $\frac{1}{2}$ is due to the existing inverse covariance matrix term in the relative entropies of orders greater than 1, which is not applicable to rank deficient matrices expected in the case of well-separated multi-modal distributions. Furthermore, the order $\frac{1}{2}$ relative entropy, well-known as the fidelity score, is a commonly-used and well-analyzed relative entropy case in quantum information theory.

To interpret the application of the order-$\frac{1}{2}$ relative entropy, we show the following theorem discussing how the negative RRKE score could approximate the joint diversity between two input multimodal distributions with sub-Gaussian components.

**Theorem 4.** *Suppose that $\mathbf{X}, \mathbf{Y} \in \mathbb{R}^d$ are random vectors with mixture distributions $P_{\text{MM}_1} = \sum_{i=1}^k \omega_i P_i$ and $P_{\text{MM}_2} = \sum_{i=1}^k \eta_i Q_i$, respectively, where $\omega_i, \eta_i$ denote the frequency of the ith component. We also assume that $P_i$ and $Q_i$ have mean vectors $\boldsymbol{\mu}_i$ and $\boldsymbol{\zeta}_i$ respectively and are both $\sigma_i$-sub-Gaussian. Then, the following approximation error bound holds for order-$\frac{1}{2}$ relative Rényi entropy where we define $\alpha_i^2 = 1 + 2\sigma_i^2/\sigma^2$:*

$$\left| \exp \left( -\text{RRKE}_2^{\text{G}_\sigma}(\mathbf{X}, \mathbf{Y}) \right) - \sum_{i=1}^k \sum_{j=1}^k \sqrt{\omega_i \eta_j} e^{-\frac{\|\boldsymbol{\mu}_i - \boldsymbol{\zeta}_j\|_2^2}{\sigma^2}} \right| \leq \sqrt[4]{\sum_{i=1}^k 32(\omega_i + \eta_i)(1 - \alpha_i^{-d})}$$

*Proof.* We defer the proof to the Appendix. □

The above theorem shows that if the $i$th mode of mixture distribution $P_{\text{mm}_1}$ and the $j$th mode of mixture distribution $P_{\text{mm}_2}$ are sufficiently close, then they add nearly $\sqrt{\omega_i \eta_j}$ to the RRKE score.

## 5 Estimation Procedure and Guarantees

As discussed earlier, the RKE and RRKE scores for a proper kernel function provide measures of the absolute and relative mode-based diversity of multi-modal distributions. Here, we propose kernel-based estimators for these entropy scores. Our estimators suggest computationally and statistically feasible ways for approximating the entropy measures by exploiting the connections between the kernel similarity values and the Rényi entropy scores. In addition, we provide non-asymptotic estimation error bounds for the proposed estimators to analyze their sample complexity. Regarding the RKE score, the following results connect this score with the kernel function $k(\mathbf{x}, \mathbf{x}')$.

**Theorem 5.** *Given a random vector $\mathbf{X}$ distributed as $P_X$, the order-2 RKE score is the result of the following equation, where $\mathbf{X}, \mathbf{X}'$ are IID draws of $P_X$:*

$$\text{RKE}_2(\mathbf{X}) = -\log \left( \mathbb{E}_{X, X' \overset{\text{iid}}{\sim} P_X} \left[ k^2(\mathbf{X}, \mathbf{X}') \right] \right).$$

**Corollary 2.** *For samples $\mathbf{x}_1, \ldots, \mathbf{x}_n$ and $K_{XX} = [\frac{1}{n} k(\mathbf{x}_i, \mathbf{x}_j)]_{n \times n}$ being the normalized kernel matrix for the observed samples, the empirical order-2 Kernel entropy is given by*

$$\widehat{\text{RKE}}_2(\mathbf{X}) = -\log \left( \|K_{XX}\|_F^2 \right),$$

*Proof.* We defer the proof to the Appendix. We note that the result of Corollary 2 for the empirical RKE score has already been shown and discussed in [14]. □

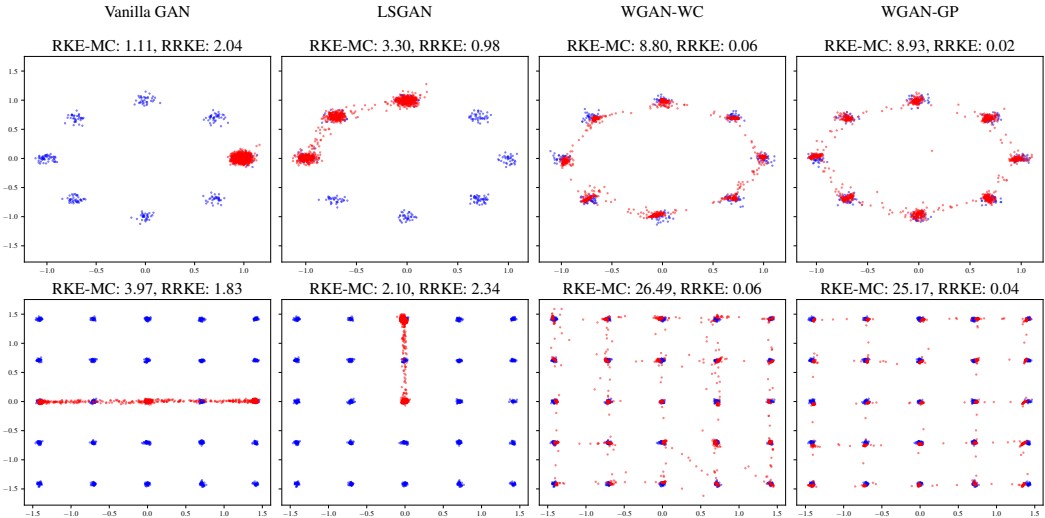

Figure 1: GANs' generated samples for Gaussian mixtures and the RKE-MC ↑ and RRKE ↓ scores.

As the above results suggest, we can use the normalized Frobenius norm of the kernel matrix to estimate the RKE score. Therefore, the computation of order-2 RKE only requires the Frobenius norm of the kernel matrix, which can be computed in $O(n^2)$ complexity. We note that for a general order-$\alpha$ Rényi entropy, one can apply the randomized algorithm in [25] for the computation of the order-$\alpha$ RKE score. Our next result bounds the estimation error for the empirical RKE score.

**Theorem 6.** *Consider a normalized kernel function satisfying $k(\mathbf{x}, \mathbf{x}) = 1$ for every $\mathbf{x} \in \mathcal{X}$. Then, for every $\delta > 0$ the following bound will hold with probability at least $1 - \delta$:*

$$\left| \exp\left(-\widehat{\mathrm{RKE}}_2(\mathbf{X})\right) - \exp\left(-\mathrm{RKE}_2(\mathbf{X})\right) \right| \leq O\left(\sqrt{\frac{\log \frac{n}{\delta}}{n}}\right)$$

*Proof.* We defer the proof to the Appendix. ∎

As implied by the above results, the order-2 Rényi kernel entropy can be efficiently estimated from training data and the probability of an $\epsilon$-large error will exponentially diminish with the sample size $n$. Next, we discuss the computation approach for the order-$\frac{1}{2}$ RRKE score. The following result reveals the kernel-based representation of this score in the empirical case.

**Theorem 7.** *Consider empirical samples $\mathbf{x}_1, \ldots, \mathbf{x}_n$ drawn from $P_X$ and $\mathbf{y}_1, \ldots, \mathbf{y}_m$ drawn from $P_Y$. Then, the following identity holds for their empirical order-$\frac{1}{2}$ RRKE score:*

$$\widehat{\mathrm{RRKE}}_{\frac{1}{2}}(\mathbf{X}, \mathbf{Y}) = -\log\left(\left\| K_{XY} \right\|_*^2\right)$$

*where $K_{XY} = \left[\frac{1}{\sqrt{nm}} k(\mathbf{x}_i, \mathbf{y}_j)\right]_{n \times m}$ denotes the normalized cross kernel matrix and $\| \cdot \|_*$ denotes the nuclear norm, i.e. the sum of a matrix's singular values.*

*Proof.* We defer the proof to the Appendix. ∎

As implied by the above theorem, the RRKE score can be computed using the singular value decomposition (SVD) for finding the singular values of $K_{XY}$. Note that the application of SVD requires $O(\min\{m^2 n, mn^2\})$ computations given $n, m$ samples from $P_X$ and $P_Y$, respectively. Therefore, computing the RRKE score could be more expensive than the RKE score, since the nuclear norm is more costly to compute than the Frobenius norm.

## 6 Numerical Results

We tested the performance of the proposed entropy-based diversity evaluation approach on several combinations of standard GAN architectures and datasets. Specifically, we used the synthetic 8-component and 25-component Gaussian mixture datasets in [26] and the following image datasets:

Table 1: Evaluated scores for three image datasets. RKE-MC (Mode Count) denotes $\exp(\mathrm{RKE})$.

| | Method | IS ↑ | FID ↓ | Precision ↑ | Recall ↑ | Density ↑ | Coverage ↑ | **RRKE ↓** | **RKE-MC ↑** |
|---|---|---|---|---|---|---|---|---|---|
| **CIFAR-10** | Dataset | 11.57 | - | - | - | - | - | - | 39.58 |
| | NVAE | 5.85 | 51.67 | 0.36 | 0.50 | 0.28 | 0.60 | 2.01 | 17.65 |
| | VDVAE | 10.51 | 37.51 | 0.34 | 0.78 | 0.23 | 0.21 | 1.81 | 32.49 |
| | DCGAN | 5.75 | 54.30 | 0.59 | 0.25 | 0.49 | 0.23 | 0.98 | 10.19 |
| | WGAN-WC | 2.59 | 157.26 | 0.36 | 0.00 | 0.18 | 0.03 | 2.09 | 10.64 |
| | WGAN-GP | 7.51 | 21.66 | 0.62 | 0.56 | 0.57 | 0.51 | 0.74 | 19.07 |
| | SAGAN | 8.62 | 10.17 | 0.68 | 0.62 | 0.73 | 0.73 | 0.65 | 24.46 |
| | SNGAN | 8.81 | 9.23 | 0.70 | 0.62 | 0.77 | 0.74 | 0.62 | 25.83 |
| | ContraGAN | 9.69 | 4.02 | 0.75 | 0.62 | 0.99 | 0.86 | 0.52 | 29.80 |
| **Tiny-ImageNet** | Dataset | 33.99 | - | - | - | - | - | - | 155.86 |
| | SAGAN | 8.21 | 46.98 | 0.55 | 0.49 | 0.44 | 0.27 | 1.42 | 25.68 |
| | SNGAN | 8.12 | 48.96 | 0.55 | 0.46 | 0.40 | 0.26 | 1.46 | 27.18 |
| | BigGAN | 11.57 | 27.34 | 0.60 | 0.58 | 0.53 | 0.43 | 1.23 | 39.61 |
| | ContraGAN | 13.79 | 21.36 | 0.54 | 0.54 | 0.54 | 0.45 | 1.26 | 56.94 |
| **ImageNet** | Dataset | 357.35 | - | - | - | - | - | - | 1823.52 |
| | SAGAN-256 | 29.67 | 44.66 | 0.57 | 0.58 | 0.42 | 0.35 | 2.34 | 105.57 |
| | SNGAN-256 | 31.92 | 35.75 | 0.54 | 0.64 | 0.41 | 0.38 | 2.22 | 115.62 |
| | ContraGAN-256 | 24.91 | 34.79 | 0.67 | 0.51 | 0.64 | 0.33 | 2.54 | 152.89 |
| | BigGAN-256 | 28.33 | 33.48 | 0.58 | 0.61 | 0.49 | 0.37 | 2.28 | 106.07 |
| | ReACGAN-256 | 52.53 | 15.65 | 0.74 | 0.42 | 0.79 | 0.41 | 2.15 | 119.76 |
| | BigGAN-2048 | 96.42 | 4.49 | 0.71 | 0.58 | 0.80 | 0.65 | 1.83 | 606.18 |
| | StyleGAN-XL | 204.73 | 1.94 | 0.77 | 0.61 | 0.67 | 0.81 | 1.50 | 1375.17 |
| | LDM-4-G | 242.62 | 3.60 | 0.86 | 0.60 | 0.69 | 0.78 | 1.56 | 1321.24 |
| | ADM-G | 188.70 | 3.86 | 0.82 | 0.64 | 0.66 | 0.82 | 1.47 | 1407.75 |

CIFAR-10 [27], Tiny-ImageNet [28], MS-COCO [29], AFHQ [30], FFHQ [31] and ImageNet [32]. We evaluated the performance of the following list of widely-used VAE, GAN and diffusion model architectures: NVAE [33], Very Deep VAE (VDVAE) [34], Vanilla GAN [1], LSGAN [35], Wassesrtein GAN with weight clipping (WGAN-WC) [18], Wassesrtein GAN with gradient penalty (WGAN-GP) [26], DCGAN [36], Self-Attention GAN (SAGAN) [37], Spectrally-Normalized GAN (SNGAN) [38], ContraGAN [39], ReACGAN [40], BigGAN [3], StyleGAN3 [41], StyleGAN-XL [42], GigaGAN [43], LDM [44] ADM-G [45] and BK-SDM [46]. To have a fair evaluation of the models, we downloaded the trained generative models from the StudioGAN repository [47].

In our evaluation of generative models, we compared the performance of order-2 Rényi Kernel Entropy Mode Count (RKE-MC), defined as $\exp(\mathrm{RKE}_2(\mathbf{X}))$, and order-$\frac{1}{2}$ Relative Rényi Kernel Entropy (RRKE) with the following standard baselines widely used in the evaluation of generative models: Inception Score (IS) [9], Fréchet Inception Distance (FID) [7], Kernel Inception Distance (KID) [8], precision and recall [11], density and coverage [12].

To compute the RKE and RRKE scores for the Gaussian mixture cases, we measured the scores based on the output of the trained generator. For image datasets, we followed the standard approach in the literature and evaluated the scores for the representation of the generator's output characterized by an Inception-net-V3 model pre-trained on ImageNet. We note that the Inception-net-based evaluation methodology is consistent with the baseline methods. Also, to select the bandwidth parameter $\sigma$ for the Gaussian kernel in the RKE and RRKE scores, we performed cross-validation and chose the smallest bandwidth $\sigma$ for which the reported score's standard deviation across 5,000 validation samples is below 0.01. Note that if the kernel bandwidth becomes overly small, the RKE score will grow almost logarithmically with the number of samples, and the standard deviation of its exponential, i.e. RKE mode count (RKE-MC), will increase almost linearly with the sample size and thus suffer from a large variance across disjoint sample sets. We provide a more detailed discussion of the bandwidth parameter's selection and the resulting variance in the Appendix.

**Diversity Evaluation for Synthetic Mixture Datasets**. We measured the RKE-MC and RRKE scores for the 8 and 25 component Gaussian mixture datasets in [26]. Figure 1 shows the real samples in blue and GANs' generated samples in red. As the evaluated scores suggest, the RKE-MC scores managed to count the number of captured modes and the RRKE relative distance increased under a

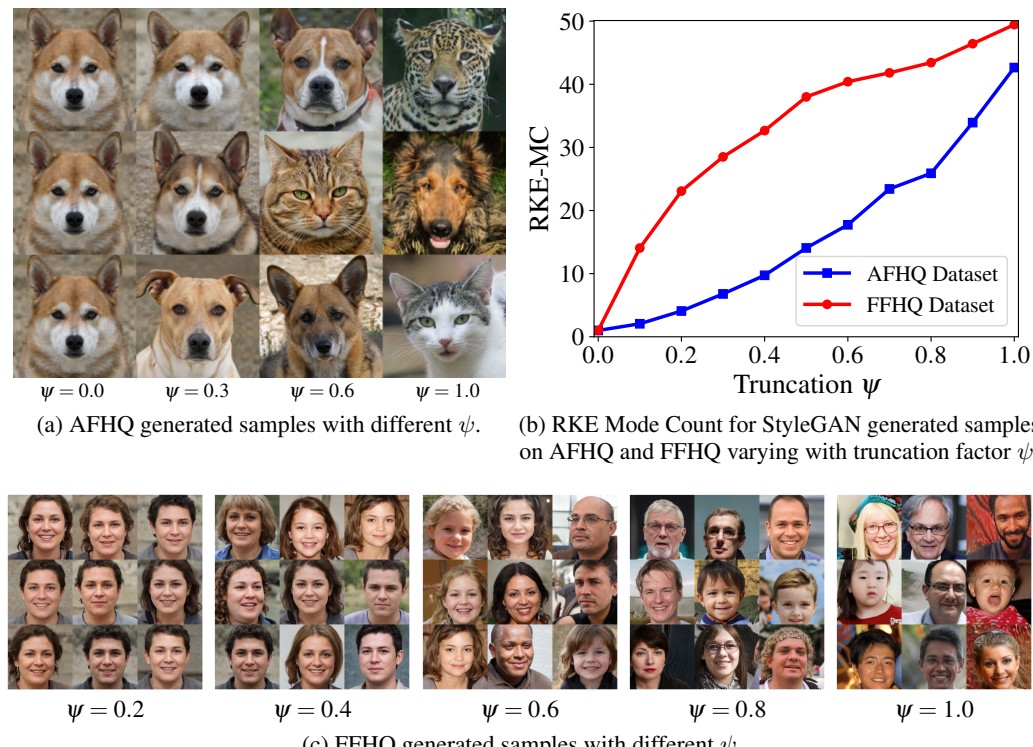

(a) AFHQ generated samples with different $\psi$.

(b) RKE Mode Count for StyleGAN generated samples on AFHQ and FFHQ varying with truncation factor $\psi$.

(c) FFHQ generated samples with different $\psi$.

Figure 2: RKE mode count's behavior under the truncation of the StyleGAN generated samples on AFHQ and FFHQ datasets.

worse coverage of the underlying Gaussian mixture. The rest of our numerical results on mixtures of Gaussians are discussed in the Appendix.

**Diversity Evaluation for Real Image Datasets**. We measured the proposed and baseline scores on the mentioned image datasets. Table 1 contains the evaluated results. Also, since the IS is a combination of both diversity and quality factors, we propose the following information-theoretic decomposition of IS to IS-quality and IS-diversity:

$$\text{IS}(X) := \exp\big(I(X;\hat{Y})\big) = \exp\big(H(\hat{Y})\big)\exp\big(-H(\hat{Y}|X)\big),$$

where we call $\exp(H(\hat{Y}))$ IS-diversity and $\exp(-H(\hat{Y}|X))$ IS-quality. The decomposed Inception and KID scores are presented in the complete table of our numerical evaluations in the Appendix.

Our numerical results show that the RKE score consistently agrees with the majority of other diversity scores, and also for all GANs remains lower than the actual dataset's RKE. Therefore, due to the estimation guarantee in Theorem 6, our numerical results provide an information-theoretic numerical proof for the empirical result in [15] that applies the birthday paradox to show GAN models cannot capture the full diversity of training data. In addition, the recent generative models StyleGAN-XL and ADM-G achieved the highest RKE, showing their diversity improvement over other generative model baselines.

Also, while the coverage and IS-diversity scores were able to differentiate between the CIFAR-10-trained generative models, WGAN-WC had the lowest score for coverage and recall, despite the Inception score reporting it as the most diverse case. Meanwhile, RKE ranked the absolute diversity of WGAN-WC to be similar to DCGAN while RRKE score shows that the modes captured by WGAN-WC are not common with that of the dataset. In the ImageNet experiments, RKE scores suggested that ContraGAN's samples are more diverse than SAGAN and SNGAN, while its coverage and recall scores were lower than those baselines. However, we note that ContraGAN reached a worse RRKE but its better RKE indicates that it captures a diverse set of modes that could have a smaller intersection with the actual ImageNet modes. The above evaluation of absolute vs. relative diversity of generated samples of ContraGAN was not revealed by the other evaluation metrics.

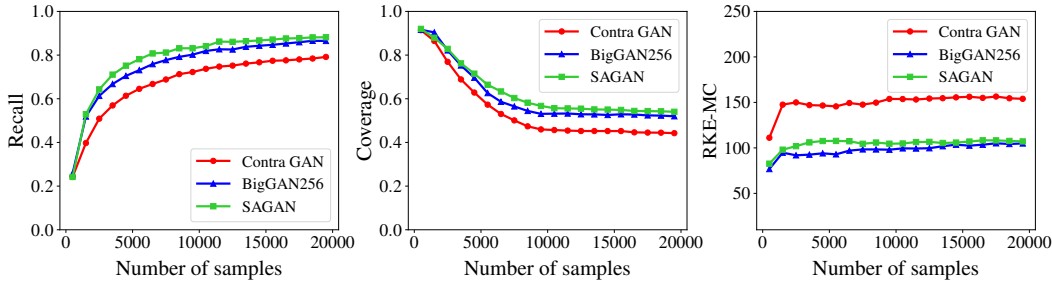

Figure 3: Comparing convergence of Recall, Coverage, and RKE scores on ImageNet dataset.

To assess the correlation between data diversity and RKE score, we repeated the dataset truncation experiment of [11, 12] for the RKE-MC measurement over AFHQ and FFHQ datasets. The numerical results in Figure 2 indicate a significant correlation between the truncation factor and the evaluated RKE. Also, we observed that the empirical RKE-MC scores manage to converge to the underlying RKE-MC using relatively few samples. Figure 3 plots the evaluated RKE-MC, recall, and coverage scores under different sample sizes, which shows RKE-MC can be estimated well with $\approx 2000$ data.

**Diversity Evaluation for text-to-image generative models.** We used the proposed RKE and RRKE scores to evaluate text-to-image generative models. In our experiments, we evaluated the state-of-the-art text-to-image generative model GigaGAN and BK-SDM on the MS-COCO dataset. The numerical results in Table 2 indicate that the GigaGAN model achieves lower RRKE in comparison to BK-SDM. On the other hand, BK-SDM reaches higher RKE-based absolute mode diversity compared to GigaGAN.

Table 2: Zero-shot evaluation on 30K images from MSCOCO validation set for text-to-image generative models. RKE-MC (Mode Count) denotes $\exp(\text{RKE})$.

| | Method | IS ↑ | FID ↓ | **RRKE ↓** | **RKE-MC ↑** |
|---|---|---|---|---|---|
| COCO | GigaGAN | 33.34 | 9.09 | 0.39 | 58.78 |
| | BK-SDM (Base) | 33.79 | 15.76 | 0.44 | 73.05 |

## 7   Conclusion

In this work, we proposed a diversity evaluation method for generative models based on entropy measures in quantum information theory. The proposed matrix-based Rényi entropy scores were shown to correlate with the number of modes in a mixture distribution with sub-Gaussian components and can be estimated from empirical data with theoretical guarantees. Our numerical results suggest that while state-of-the-art generative models reach a lower entropy-based diversity score than training data, the recent GAN and diffusion model architectures such as StyleGAN-XL and ADM manage to significantly improve the diversity factor over earlier geneartive models. A future direction for our work is to extend the diversity evaluation to non-GAN and non-diffusion models such as variational autoencoders (VAEs) and flow-based models. Also, studying the effects of the pre-trained Inception model on RKE and RRKE evaluations and comparing their robustness to the choice of pre-trained models vs. the baseline scores studied in [48] will be an interesting topic for future exploration.

## Acknowledgments

The work of Farzan Farnia was partially supported by a grant from the Research Grants Council of the Hong Kong Special Administrative Region, China, Project 14209920 and by a CUHK Direct Research Grant [CUHK Project No. 4055164]. The work of Cheuk Ting Li was partially supported by an ECS grant from the Research Grants Council of the Hong Kong Special Administrative Region, China [Project No.: CUHK 24205621]. The authors also thank the anonymous reviewers and metareviewer for their constructive feedback and suggestions.

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
