# A Appendix

## A.1 Proof of Proposition 1

The proposition is directly implied by the fact that the eigenvalues of $XX^\top = X^2$ are $\lambda_1^2, \dots, \lambda_d^2$. Therefore, we have

$$\|X\|_F^2 = \mathrm{Tr}(XX^\top) = \mathrm{Tr}(X^2) = \sum_{i=1}^d \lambda_i^2.$$

The proof is therefore a direct consequence of the definition of order-2 Rényi entropy.

## A.2 Proof of Theorem 1

We apply Theorem 5 which reveals that for a Gaussian kernel bandwidth of $\sqrt{2}\sigma$ the following holds. Note that, without loss of generality and for simplicity of theoretical derivations, we derive the equations for a bandwidth of $\sqrt{2}\sigma$:

$$\mathrm{RKE}_2^{G_{\sqrt{2}\sigma}}(\mathbf{X}) = \mathbb{E}_{\mathbf{X},\mathbf{X}'\sim P_X}\left[k_{\sqrt{2}\sigma}^2(\mathbf{X},\mathbf{X}')\right]$$

$$= \sum_{i=1}^k \sum_{j=1}^k \omega_i \omega_j \mathbb{E}_{\mathbf{X}\sim\mathcal{N}(\boldsymbol{\mu}_i,\Sigma_i), X'\sim\mathcal{N}(\boldsymbol{\mu}_j,\Sigma_j)}\left[k_{\sqrt{2}\sigma}^2(\mathbf{X},\mathbf{X}')\right]$$

$$= \sum_{i=1}^k \sum_{j=1}^k \omega_i \omega_j \int \frac{-1}{\sqrt{(2\pi)^{2d}\det(\Sigma_i)\det(\Sigma_j)}}$$
$$\times \exp\left(\frac{-1}{2}\left(\|\mathbf{x}-\boldsymbol{\mu}_i\|_{\Sigma_i^{-1}}^2 + \|\mathbf{x}'-\boldsymbol{\mu}_j\|_{\Sigma_j^{-1}}^2 + \sigma^{-2}\|\mathbf{x}-\mathbf{x}'\|_2^2\right)\right)\mathrm{d}\mathbf{x}\mathrm{d}\mathbf{x}'$$

$$= \sum_{i=1}^k \sum_{j=1}^k \omega_i \omega_j \int \frac{-1}{\sqrt{(2\pi)^{2d}\det(\Sigma_i)\det(\Sigma_j)}} \times \exp\left(\frac{-1}{2}\left(\|\mathbf{x}-\boldsymbol{\mu}_i\|_{\Sigma_i^{-1}}^2\right.\right.$$
$$\left.\left. + \|\mathbf{x}'-\boldsymbol{\mu}_j\|_{\Sigma_j^{-1}}^2 + \sigma^{-2}\|(\mathbf{x}-\boldsymbol{\mu}_i)-(\mathbf{x}'-\boldsymbol{\mu}_j)+(\boldsymbol{\mu}_i-\boldsymbol{\mu}_j)\|_2^2\right)\right)\mathrm{d}\mathbf{x}\mathrm{d}\mathbf{x}'$$

$$= \sum_{i=1}^k \sum_{j=1}^k \omega_i \omega_j \int \frac{1}{\sqrt{(2\pi)^{2d}\det(\Sigma_i\Sigma_j)}}$$
$$\times \exp\left(\frac{-1}{2}\begin{bmatrix}\mathbf{x}-\boldsymbol{\mu}_i\\\mathbf{x}'-\boldsymbol{\mu}_j\end{bmatrix}^\top \begin{bmatrix}\Sigma_i^{-1}+\sigma^{-2}I & -\sigma^{-2}I\\ -\sigma^{-2}I & \Sigma_j^{-1}+\sigma^{-2}I\end{bmatrix}\begin{bmatrix}\mathbf{x}-\boldsymbol{\mu}_i\\\mathbf{x}'-\boldsymbol{\mu}_j\end{bmatrix}\right.$$
$$\left. + \begin{bmatrix}\mathbf{x}-\boldsymbol{\mu}_i\\\mathbf{x}'-\boldsymbol{\mu}_j\end{bmatrix}^\top \begin{bmatrix}\frac{1}{\sigma^2}(\boldsymbol{\mu}_j-\boldsymbol{\mu}_i)\\\frac{1}{\sigma^2}(\boldsymbol{\mu}_i-\boldsymbol{\mu}_j)\end{bmatrix} - \frac{1}{2\sigma^2}\|\boldsymbol{\mu}_i-\boldsymbol{\mu}_j\|_2^2\right)\mathrm{d}\mathbf{x}\mathrm{d}\mathbf{x}'$$

$$\overset{(a)}{=} \sum_{i=1}^k \sum_{j=1}^k \omega_i \omega_j \int \frac{1}{\sqrt{(2\pi)^{2d}\det(\Sigma_i\Sigma_j)}} \exp\left(\frac{-1}{2}\left(\begin{bmatrix}\mathbf{x}-\boldsymbol{\mu}_i\\\mathbf{x}'-\boldsymbol{\mu}_j\end{bmatrix}-\mathbf{b}_{i,j}\right)^\top C_{i,j}\left(\begin{bmatrix}\mathbf{x}-\boldsymbol{\mu}_i\\\mathbf{x}'-\boldsymbol{\mu}_j\end{bmatrix}-\mathbf{b}_{i,j}\right)\right.$$
$$\left. + \frac{1}{2}\mathbf{b}_{i,j}^\top C_{i,j}\mathbf{b}_{i,j} - \frac{1}{2\sigma^2}\|\boldsymbol{\mu}_i-\boldsymbol{\mu}_j\|_2^2\right)\mathrm{d}\mathbf{x}\mathrm{d}\mathbf{x}'$$

$$= \sum_{i=1}^k \sum_{j=1}^k \omega_i \omega_j \exp\left(\frac{1}{2}\mathbf{b}_{i,j}^\top C_{i,j}\mathbf{b}_{i,j} - \frac{1}{2\sigma^2}\|\boldsymbol{\mu}_i-\boldsymbol{\mu}_j\|_2^2\right)\int \frac{1}{\sqrt{(2\pi)^{2d}\det(\Sigma_i\Sigma_j)}}$$
$$\times \exp\left(\frac{-1}{2}\left(\begin{bmatrix}\mathbf{x}-\boldsymbol{\mu}_i\\\mathbf{x}'-\boldsymbol{\mu}_j\end{bmatrix}-\mathbf{b}_{i,j}\right)^\top C_{i,j}\left(\begin{bmatrix}\mathbf{x}-\boldsymbol{\mu}_i\\\mathbf{x}'-\boldsymbol{\mu}_j\end{bmatrix}-\mathbf{b}_{i,j}\right)\right)\mathrm{d}\mathbf{x}\mathrm{d}\mathbf{x}'$$

$$\overset{(b)}{=} \sum_{i=1}^k \sum_{j=1}^k \omega_i \omega_j \exp\left(\frac{1}{2}\mathbf{b}_{i,j}^\top C_{i,j}\mathbf{b}_{i,j} - \frac{1}{2\sigma^2}\|\boldsymbol{\mu}_i-\boldsymbol{\mu}_j\|_2^2\right)\frac{\sqrt{(2\pi)^{2d}\det(C_{i,j}^{-1})}}{\sqrt{(2\pi)^{2d}\det(\Sigma_i\Sigma_j)}}$$

$$\overset{(c)}{=} \sum_{i=1}^{k}\sum_{j=1}^{k} \omega_i\omega_j \exp\Big(\frac{1}{2}\mathbf{b}_{i,j}^\top C_{i,j}\mathbf{b}_{i,j} - \frac{1}{2\sigma^2}\|\boldsymbol{\mu}_i - \boldsymbol{\mu}_j\|_2^2\Big) \frac{1}{\sqrt{\det(\begin{bmatrix}\Sigma_i & 0 \\ 0 & \Sigma_j\end{bmatrix}C_{i,j})}}$$

$$= \sum_{i=1}^{k}\sum_{j=1}^{k} \omega_i\omega_j \exp\Big(\frac{1}{2}\mathbf{b}_{i,j}^\top C_{i,j}\mathbf{b}_{i,j} - \frac{1}{2\sigma^2}\|\boldsymbol{\mu}_i - \boldsymbol{\mu}_j\|_2^2\Big) \frac{1}{\sqrt{\det(\begin{bmatrix}I+\sigma^{-2}\Sigma_i & -\sigma^{-2}\Sigma_i \\ -\sigma^{-2}\Sigma_j & I+\sigma^{-2}\Sigma_j\end{bmatrix})}}$$

$$\overset{(d)}{=} \sum_{i=1}^{k}\sum_{j=1}^{k}\Big[\omega_i\omega_j \exp\Big(\frac{1}{2\sigma^2}(\boldsymbol{\mu}_i - \boldsymbol{\mu}_j)^\top\big((I-A_{i,j})-I\big)(\boldsymbol{\mu}_i - \boldsymbol{\mu}_j)\Big)$$

$$\times \det\Big(I + \frac{1}{\sigma^2}(\Sigma_i + \Sigma_j)\Big)^{-1/2}\Big]$$

$$= \sum_{i=1}^{k}\sum_{j=1}^{k}\Big[\omega_i\omega_j \exp\Big(-\frac{1}{2\sigma^2}(\boldsymbol{\mu}_i - \boldsymbol{\mu}_j)^\top A_{i,j}(\boldsymbol{\mu}_i - \boldsymbol{\mu}_j)\Big) \det\Big(I + \frac{1}{\sigma^2}(\Sigma_i + \Sigma_j)\Big)^{-1/2}\Big]$$

$$= \sum_{i=1}^{k}\sum_{j=1}^{k}\Big[\omega_i\omega_j e^{-\frac{\|\boldsymbol{\mu}_i-\boldsymbol{\mu}_j\|_{A_{i,j}}^2}{2\sigma^2}} \det\Big(I + \frac{1}{\sigma^2}(\Sigma_i + \Sigma_j)\Big)^{-1/2}\Big],$$

Here $(a)$ comes from defining $C_{i,j} = \begin{bmatrix}\Sigma_i^{-1}+\sigma^{-2}I & -\sigma^{-2}I \\ -\sigma^{-2}I & \Sigma_j^{-1}+\sigma^{-2}I\end{bmatrix}$ and $\mathbf{b}_{i,j} = \frac{1}{\sigma^2}C_{i,j}^{-1}\begin{bmatrix}\boldsymbol{\mu}_j-\boldsymbol{\mu}_i \\ \boldsymbol{\mu}_i-\boldsymbol{\mu}_j\end{bmatrix}$.
$(b)$ follows from the unit integral of a multivariate Gaussian probability density function with mean vector $\mathbf{b}_{i,j}$ and covariance matrix $C_{i,j}^{-1}$. $(c)$ uses the determinant of Block diagonal matrices as $\det(\begin{bmatrix}\Sigma_i & 0 \\ 0 & \Sigma_j\end{bmatrix}) = \det(\Sigma_i)\det(\Sigma_j) = \det(\Sigma_i\Sigma_j)$. $(d)$ is based on the determinant of block matrices $\begin{bmatrix}A & B \\ C & D\end{bmatrix}$ that if $CD = DC$ (which holds in our case as $-\sigma^{-2}\Sigma_j$ and $I+\sigma^{-2}\Sigma_j$ commute) then $\det(\begin{bmatrix}A & B \\ C & D\end{bmatrix}) = \det(AD-BC)$, and furthermore the fact that by defining $F_i = \sigma^2\Sigma_i^{-1}$ and $F_j = \sigma^2\Sigma_j^{-1}$ we will have

$$\mathbf{b}_{i,j}^\top C_{i,j}\mathbf{b}_{i,j} = \frac{1}{\sigma^4}\begin{bmatrix}\boldsymbol{\mu}_j-\boldsymbol{\mu}_i \\ \boldsymbol{\mu}_i-\boldsymbol{\mu}_j\end{bmatrix}^\top C_{i,j}^{-1}\begin{bmatrix}\boldsymbol{\mu}_j-\boldsymbol{\mu}_i \\ \boldsymbol{\mu}_i-\boldsymbol{\mu}_j\end{bmatrix}$$

$$= \frac{1}{\sigma^2}\begin{bmatrix}\boldsymbol{\mu}_j-\boldsymbol{\mu}_i \\ \boldsymbol{\mu}_i-\boldsymbol{\mu}_j\end{bmatrix}^\top (\sigma^2 C_{i,j})^{-1}\begin{bmatrix}\boldsymbol{\mu}_j-\boldsymbol{\mu}_i \\ \boldsymbol{\mu}_i-\boldsymbol{\mu}_j\end{bmatrix}$$

$$= \frac{1}{\sigma^2}\begin{bmatrix}\boldsymbol{\mu}_j-\boldsymbol{\mu}_i \\ \boldsymbol{\mu}_i-\boldsymbol{\mu}_j\end{bmatrix}^\top \begin{bmatrix}I+\sigma^2\Sigma_i^{-1} & -I \\ -I & I+\sigma^2\Sigma_j^{-1}\end{bmatrix}^{-1}\begin{bmatrix}\boldsymbol{\mu}_j-\boldsymbol{\mu}_i \\ \boldsymbol{\mu}_i-\boldsymbol{\mu}_j\end{bmatrix}$$

$$= \frac{1}{\sigma^2}\begin{bmatrix}\boldsymbol{\mu}_j-\boldsymbol{\mu}_i \\ \boldsymbol{\mu}_i-\boldsymbol{\mu}_j\end{bmatrix}^\top \begin{bmatrix}I+F_i & -I \\ -I & I+F_j\end{bmatrix}^{-1}\begin{bmatrix}\boldsymbol{\mu}_j-\boldsymbol{\mu}_i \\ \boldsymbol{\mu}_i-\boldsymbol{\mu}_j\end{bmatrix}$$

$$\overset{(e)}{=} \frac{1}{\sigma^2}\begin{bmatrix}\boldsymbol{\mu}_j-\boldsymbol{\mu}_i \\ \boldsymbol{\mu}_i-\boldsymbol{\mu}_j\end{bmatrix}^\top \begin{bmatrix}(I+F_i-(I+F_j)^{-1})^{-1} & (F_i+F_j+F_iF_j)^{-1} \\ (F_i+F_j+F_jF_i)^{-1} & (I+F_j-(I+F_i)^{-1})^{-1}\end{bmatrix}\begin{bmatrix}\boldsymbol{\mu}_j-\boldsymbol{\mu}_i \\ \boldsymbol{\mu}_i-\boldsymbol{\mu}_j\end{bmatrix}$$

$$= \frac{1}{\sigma^2}(\boldsymbol{\mu}_i-\boldsymbol{\mu}_j)^\top\Big((I+F_i-(I+F_j)^{-1})^{-1}$$

$$+ (I+F_j-(I+F_i)^{-1})^{-1} - (F_i+F_j+F_iF_j)^{-1} - (F_i+F_j+F_jF_i)^{-1}\Big)(\boldsymbol{\mu}_i-\boldsymbol{\mu}_j)$$

$$= \frac{1}{\sigma^2}(\boldsymbol{\mu}_i-\boldsymbol{\mu}_j)^\top\Big((I+\sigma^2\Sigma_i^{-1}-(I+\sigma^2\Sigma_j^{-1})^{-1})^{-1} + (I+\sigma^2\Sigma_j^{-1}-(I+\sigma^2\Sigma_i^{-1})^{-1})^{-1}$$

$$- (\sigma^2 \Sigma_i^{-1} + \sigma^2 \Sigma_j^{-1} + \sigma^4 \Sigma_i^{-1} \Sigma_j^{-1})^{-1} - (\sigma^2 \Sigma_i^{-1} + \sigma^2 \Sigma_j^{-1} + \sigma^4 \Sigma_j^{-1} \Sigma_i^{-1})^{-1} \Big) \Big( \boldsymbol{\mu}_i - \boldsymbol{\mu}_j \Big)$$

$$= \frac{1}{\sigma^2} \Big( \boldsymbol{\mu}_i - \boldsymbol{\mu}_j \Big)^\top \Big( I - A_{i,j} \Big) \Big( \boldsymbol{\mu}_i - \boldsymbol{\mu}_j \Big)$$

In the above, $(e)$ holds because $F_i$ and $F_j$ are positive definite matrices for the supposed invertible and thus positive definite $\Sigma_i$ and $\Sigma_j$. Hence, the matrices $I + F_i$ and $I + F_j$ are both positive definite and invertible. Also, the Schur complement $(I + F_j) - (-I)(I + F_i)^{-1}(-I) = I + F_j - (I + F_i)^{-1}$ will be a positive definite and invertible matrix because $(I + F_i)^{-1} \prec I \prec I + F_j$. Therefore, for the inverse of the block matrix, we will have

$$\begin{bmatrix} I + F_i & -I \\ -I & I + F_j \end{bmatrix}^{-1} = \begin{bmatrix} \big(I + F_i - (I + F_j)^{-1}\big)^{-1} & (F_i + F_j + F_i F_j)^{-1} \\ (F_i + F_j + F_j F_i)^{-1} & \big(I + F_j - (I + F_i)^{-1}\big)^{-1} \end{bmatrix}.$$

The above discussion completes the proof.

### A.3 Proof of Theorem 2

Note that according to the definition of the kernel covariance matrix we have

$$C_X - \sum_{i=1}^k \omega_i \phi(\boldsymbol{\mu}_i) \phi(\boldsymbol{\mu}_i)^\top = \sum_{i=1}^k \Big[ \omega_i \Big( \mathbb{E}\big[\phi(\boldsymbol{\mu}_i + Z_i) \phi(\boldsymbol{\mu}_i + Z_i)^\top\big] - \phi(\boldsymbol{\mu}_i) \phi(\boldsymbol{\mu}_i)^\top \Big) \Big].$$

Therefore, applying Jensen's inequality for the convex Frobenius norm-squared function $\| \cdot \|_F^2$ shows that

$$\Big\| C_X - \sum_{i=1}^k \omega_i \phi(\boldsymbol{\mu}_i) \phi(\boldsymbol{\mu}_i)^\top \Big\|_F^2$$

$$= \Big\| \sum_{i=1}^k \Big[ \omega_i \Big( \mathbb{E}\big[\phi(\boldsymbol{\mu}_i + Z_i) \phi(\boldsymbol{\mu}_i + Z_i)^\top\big] - \phi(\boldsymbol{\mu}_i) \phi(\boldsymbol{\mu}_i)^\top \Big) \Big] \Big\|_F^2$$

$$\leq \sum_{i=1}^k \Big[ \omega_i \Big\| \mathbb{E}\big[\phi(\boldsymbol{\mu}_i + Z_i) \phi(\boldsymbol{\mu}_i + Z_i)^\top\big] - \phi(\boldsymbol{\mu}_i) \phi(\boldsymbol{\mu}_i)^\top \Big\|_F^2 \Big]$$

$$= \sum_{i=1}^k \Big[ \omega_i \Big\| \mathbb{E}\big[\phi(\boldsymbol{\mu}_i + Z_i) \phi(\boldsymbol{\mu}_i + Z_i)^\top - \phi(\boldsymbol{\mu}_i) \phi(\boldsymbol{\mu}_i)^\top\big] \Big\|_F^2 \Big]$$

$$\leq \sum_{i=1}^k \omega_i \mathbb{E}\Big[\big\| \phi(\boldsymbol{\mu}_i + Z_i) \phi(\boldsymbol{\mu}_i + Z_i)^\top - \phi(\boldsymbol{\mu}_i) \phi(\boldsymbol{\mu}_i)^\top \big\|_F^2\Big]$$

$$= \sum_{i=1}^k \omega_i \mathbb{E}\Big[2 - 2\big(\phi(\boldsymbol{\mu}_i)^\top \phi(\boldsymbol{\mu}_i + Z_i)\big)^2\Big]$$

$$= \sum_{i=1}^k 2\omega_i \mathbb{E}\Big[1 - \exp\big(\frac{\|\mathbf{Z}_i\|_2^2}{\sigma^2}\big)\Big]$$

$$\leq \sum_{i=1}^k 2\omega_i \big(1 - \frac{1}{\alpha_i^d}\big).$$

Note that the inequality before the last holds, since $\phi(\boldsymbol{\mu}_i + Z_i) \phi(\boldsymbol{\mu}_i + Z_i)^\top - \phi(\boldsymbol{\mu}_i) \phi(\boldsymbol{\mu}_i)^\top$ is a rank two matrix where $\mathbf{a} = \phi(\boldsymbol{\mu}_i + Z_i)$ and $\mathbf{b} = \phi(\boldsymbol{\mu}_i)$ have both unit norms. Therefore, for the Frobenius norm-squared of $\mathbf{a}\mathbf{a}^\top - \mathbf{b}\mathbf{b}^\top$ will be equal to $2 - 2(\mathbf{a}^\top \mathbf{b})^2$. Therefore, we have

$$\Big| \|C_X\|_F - \|\sum_{i=1}^k \omega_i \phi(\boldsymbol{\mu}_i) \phi(\boldsymbol{\mu}_i)^\top\|_F \Big| \leq \Big\| C_X - \sum_{i=1}^k \omega_i \phi(\boldsymbol{\mu}_i) \phi(\boldsymbol{\mu}_i)^\top \Big\|_F \leq \sqrt{\sum_{i=1}^k 2\omega_i \big(1 - \frac{1}{\alpha_i^d}\big)}.$$

On the other hand, the special zero-covariance case of Theorem 1 shows that

$$\Big\|\sum_{i=1}^{k} \omega_i \phi(\boldsymbol{\mu}_i)\phi(\boldsymbol{\mu}_i)^{\top}\Big\|_F^2 = \sum_{i=1}^{k}\sum_{j=1}^{k} \omega_i\omega_j \exp\big(-\frac{\|\boldsymbol{\mu}_i - \boldsymbol{\mu}_j\|_2^2}{\sigma^2}\big)$$

Also, we know that the Gaussian kernel is always upper-bounded by 1 and thus $\big\|C_X\big\|_F + \big\|\sum_{i=1}^{k}\omega_i\phi(\boldsymbol{\mu}_i)\phi(\boldsymbol{\mu}_i)^{\top}\big\|_F \le 2$. Therefore, knowing that $\mathrm{RKE}_2^{G_\sigma}(\mathbf{X}) = -\log\big(\|C_X\|_F^2\big)$ shows

$$\Big|\exp\big(-\mathrm{RKE}_2^{G_\sigma}(\mathbf{X})\big) - \sum_{i=1}^{k}\sum_{j=1}^{k}\omega_i\omega_j\exp\big(-\frac{\|\boldsymbol{\mu}_i - \boldsymbol{\mu}_j\|_2^2}{\sigma^2}\big)\Big|$$

$$= \Big|\big\|C_X\big\|_F^2 - \sum_{i=1}^{k}\sum_{j=1}^{k}\omega_i\omega_j\exp\big(-\frac{\|\boldsymbol{\mu}_i - \boldsymbol{\mu}_j\|_2^2}{\sigma^2}\big)\Big|$$

$$= \Big|\big\|C_X\big\|_F^2 - \big\|\sum_{i=1}^{k}\omega_i\phi(\boldsymbol{\mu}_i)\phi(\boldsymbol{\mu}_i)^{\top}\big\|_F^2\Big|$$

$$= \Big|\big\|C_X\big\|_F + \big\|\sum_{i=1}^{k}\omega_i\phi(\boldsymbol{\mu}_i)\phi(\boldsymbol{\mu}_i)^{\top}\big\|_F\Big| \times \Big|\big\|C_X\big\|_F - \big\|\sum_{i=1}^{k}\omega_i\phi(\boldsymbol{\mu}_i)\phi(\boldsymbol{\mu}_i)^{\top}\big\|_F\Big|$$

$$\le 2\Big\|C_X - \sum_{i=1}^{k}\omega_i\phi(\boldsymbol{\mu}_i)\phi(\boldsymbol{\mu}_i)^{\top}\Big\|_F$$

$$\le \sqrt{\sum_{i=1}^{k} 8\omega_i\big(1 - \frac{1}{\alpha_i^d}\big)}.$$

The theorem's proof is hence complete.

### A.4 Proof of Theorem 3

By the continuity of $P_X$, we have

$$\lim_{\sigma\to 0}\big(\pi\sigma^2\big)^{-d/2}\,\mathbb{E}\big[k_\sigma^2(X, X')\big]$$

$$= \lim_{\sigma\to 0}\big(\pi\sigma^2\big)^{-d/2}\,\mathbb{E}\Big[\int k_\sigma^2(X, x')P_X(x')\mathrm{d}x'\Big]$$

$$= \lim_{\sigma\to 0}\mathbb{E}\Big[\int \big(\pi\sigma^2\big)^{-d/2}\exp\Big(-\frac{\|X - x'\|_2^2}{\sigma^2}\Big)P_X(x')\mathrm{d}x'\Big]$$

$$= \mathbb{E}\Big[\lim_{\sigma\to 0}\int \big(\pi\sigma^2\big)^{-d/2}\exp\Big(-\frac{\|X - x'\|_2^2}{\sigma^2}\Big)P_X(x')\mathrm{d}x'\Big]$$

$$= \mathbb{E}\big[P_X(X)\big]$$

$$= \int P_X(x)^2\mathrm{d}x.$$

Therefore,

$$\lim_{\sigma\to 0}\Big(\mathrm{RKE}_2(P_X) + \frac{d}{2}\log(\pi\sigma^2)\Big)$$

$$= \lim_{\sigma\to 0}\Big(-\log\Big(\big(\pi\sigma^2\big)^{-d/2}\,\mathbb{E}\big[k_\sigma^2(X, X')\big]\Big)\Big)$$

$$= -\log\Big(\int P_X(x)^2\mathrm{d}x\Big).$$

## A.5 Proof of Theorem 4

Using $\|\cdot\|_*$ to denote the nuclear norm, the following holds for every two PSD matrices $A, B \in \mathbb{R}^{d \times d}$ [24]:

$$\mathrm{Tr}\left(\sqrt{A^{1/2} B A^{1/2}}\right) = \mathrm{Tr}\left(\sqrt{A^{1/2} B^{1/2} \left(A^{1/2} B^{1/2}\right)^\top}\right) = \left\|\sqrt{A}\sqrt{B}\right\|_*. \tag{1}$$

Therefore, we can write

$$\left| \mathrm{Tr}\left(\sqrt{\sqrt{C_X} C_Y \sqrt{C_X}}\right) - \mathrm{Tr}\left(\sqrt{\sqrt{\sum_{i=1}^k \omega_i \phi(\boldsymbol{\mu}_i)\phi(\boldsymbol{\mu}_i)^\top}\left(\sum_{i=1}^k \eta_i \phi(\boldsymbol{\zeta}_i)\phi(\boldsymbol{\zeta}_i)^\top\right)\sqrt{\sum_{i=1}^k \omega_i \phi(\boldsymbol{\mu}_i)\phi(\boldsymbol{\mu}_i)^\top}}\right) \right|$$

$$\overset{(a)}{=} \left| \left\|\sqrt{C_X}\sqrt{C_Y}\right\|_* - \left\|\sqrt{\sum_{i=1}^k \omega_i \phi(\boldsymbol{\mu}_i)\phi(\boldsymbol{\mu}_i)^\top}\sqrt{\sum_{i=1}^k \eta_i \phi(\boldsymbol{\zeta}_i)\phi(\boldsymbol{\zeta}_i)^\top}\right\|_* \right|$$

$$\overset{(b)}{\leq} \left| \left\|\sqrt{C_X}\sqrt{C_Y}\right\|_* - \left\|\sqrt{C_X}\sqrt{\sum_{i=1}^k \eta_i \phi(\boldsymbol{\zeta}_i)\phi(\boldsymbol{\zeta}_i)^\top}\right\|_* \right|$$

$$+ \left| \left\|\sqrt{C_X}\sqrt{\sum_{i=1}^k \eta_i \phi(\boldsymbol{\zeta}_i)\phi(\boldsymbol{\zeta}_i)^\top}\right\|_* - \left\|\sqrt{\sum_{i=1}^k \omega_i \phi(\boldsymbol{\mu}_i)\phi(\boldsymbol{\mu}_i)^\top}\sqrt{\sum_{i=1}^k \eta_i \phi(\boldsymbol{\zeta}_i)\phi(\boldsymbol{\zeta}_i)^\top}\right\|_* \right|$$

$$\overset{(c)}{\leq} \left\| \sqrt{C_X}\left(\sqrt{C_Y} - \sqrt{\sum_{i=1}^k \eta_i \phi(\boldsymbol{\zeta}_i)\phi(\boldsymbol{\zeta}_i)^\top}\right) \right\|_*$$

$$+ \left\| \left(\sqrt{C_X} - \sqrt{\sum_{i=1}^k \omega_i \phi(\boldsymbol{\mu}_i)\phi(\boldsymbol{\mu}_i)^\top}\right)\sqrt{\sum_{i=1}^k \eta_i \phi(\boldsymbol{\zeta}_i)\phi(\boldsymbol{\zeta}_i)^\top} \right\|_*$$

$$\overset{(d)}{\leq} \left\|\sqrt{C_X}\right\|_F \left\|\sqrt{C_Y} - \sqrt{\sum_{i=1}^k \eta_i \phi(\boldsymbol{\zeta}_i)\phi(\boldsymbol{\zeta}_i)^\top}\right\|_F$$

$$+ \left\|\sqrt{C_X} - \sqrt{\sum_{i=1}^k \omega_i \phi(\boldsymbol{\mu}_i)\phi(\boldsymbol{\mu}_i)^\top}\right\|_F \left\|\sqrt{\sum_{i=1}^k \eta_i \phi(\boldsymbol{\zeta}_i)\phi(\boldsymbol{\zeta}_i)^\top}\right\|_F$$

$$\overset{(e)}{=} \left\|\sqrt{C_Y} - \sqrt{\sum_{i=1}^k \eta_i \phi(\boldsymbol{\zeta}_i)\phi(\boldsymbol{\zeta}_i)^\top}\right\|_F + \left\|\sqrt{C_X} - \sqrt{\sum_{i=1}^k \omega_i \phi(\boldsymbol{\mu}_i)\phi(\boldsymbol{\mu}_i)^\top}\right\|_F$$

$$\overset{(f)}{\leq} \sqrt{\left\|C_Y - \sum_{i=1}^k \eta_i \phi(\boldsymbol{\zeta}_i)\phi(\boldsymbol{\zeta}_i)^\top\right\|_*} + \sqrt{\left\|C_X - \sum_{i=1}^k \omega_i \phi(\boldsymbol{\mu}_i)\phi(\boldsymbol{\mu}_i)^\top\right\|_*}$$

In the above, (a) holds due to the identity discussed in (1). (b) and (c) follow from the application of triangle inequality for the absolute value and nuclear norm functions, respectively. (d) is an application of Holder's inequality for Schatten norms where $\|AB\|_* \leq \|A\|_F \|B\|_F$ for every pair of matrices $A, B$. (e) holds because all the four matrices $\sqrt{C_X}, \sqrt{C_Y}, \sqrt{\sum_{i=1}^k \omega_i \phi(\boldsymbol{\mu}_i)\phi(\boldsymbol{\mu}_i)^\top}$, $\sqrt{\sum_{i=1}^k \eta_i \phi(\boldsymbol{\zeta}_i)\phi(\boldsymbol{\zeta}_i)^\top}$ have a unit Frobenius norm since the square of their eigenvalues will be the eigenvalues of a normalized kernel covariance matrix. Finally, (f) is the result of the matrix norm inequality that $\|\sqrt{A} - \sqrt{B}\|_F \leq \sqrt{\|A - B\|_*}$ for every pair of PSD matrices $A, B$.

To further simplify the above upper-bound Next, we apply the Jensen's inequality for the convex nuclear norm-squared function which shows that

$$\left\| C_X - \sum_{i=1}^{k} \omega_i \phi(\boldsymbol{\mu}_i)\phi(\boldsymbol{\mu}_i)^\top \right\|_*^2$$

$$= \left\| \sum_{i=1}^{k} \left[ \omega_i \Big( \mathbb{E}\big[\phi(\boldsymbol{\mu}_i + Z_i)\phi(\boldsymbol{\mu}_i + Z_i)^\top\big] - \phi(\boldsymbol{\mu}_i)\phi(\boldsymbol{\mu}_i)^\top \Big) \right] \right\|_*^2$$

$$\leq \sum_{i=1}^{k} \left[ \omega_i \left\| \mathbb{E}\big[\phi(\boldsymbol{\mu}_i + Z_i)\phi(\boldsymbol{\mu}_i + Z_i)^\top\big] - \phi(\boldsymbol{\mu}_i)\phi(\boldsymbol{\mu}_i)^\top \right\|_*^2 \right]$$

$$= \sum_{i=1}^{k} \left[ \omega_i \left\| \mathbb{E}\big[\phi(\boldsymbol{\mu}_i + Z_i)\phi(\boldsymbol{\mu}_i + Z_i)^\top - \phi(\boldsymbol{\mu}_i)\phi(\boldsymbol{\mu}_i)^\top\big] \right\|_*^2 \right]$$

$$\leq \sum_{i=1}^{k} \left[ \omega_i \mathbb{E}\left[ \left\| \phi(\boldsymbol{\mu}_i + Z_i)\phi(\boldsymbol{\mu}_i + Z_i)^\top - \phi(\boldsymbol{\mu}_i)\phi(\boldsymbol{\mu}_i)^\top \right\|_*^2 \right] \right]$$

$$\leq \sum_{i=1}^{k} \left[ \omega_i \mathbb{E}\left[ 4 - 4\big(\phi(\boldsymbol{\mu}_i)^\top \phi(\boldsymbol{\mu}_i + Z_i)\big)^2 \right] \right]$$

$$= \sum_{i=1}^{k} 4\omega_i \mathbb{E}\left[ 1 - \exp\big( \frac{\|\mathbf{Z}_i\|_2^2}{\sigma^2} \big) \right]$$

$$\leq \sum_{i=1}^{k} 4\omega_i \big( 1 - \frac{1}{\alpha_i^d} \big).$$

The inequality before the last holds because $\phi(\boldsymbol{\mu}_i + Z_i)\phi(\boldsymbol{\mu}_i + Z_i)^\top - \phi(\boldsymbol{\mu}_i)\phi(\boldsymbol{\mu}_i)^\top$ is a rank two matrix with Frobenius norm-squared of $2 - 2(\phi(\boldsymbol{\mu}_i + Z_i)^\top \phi(\boldsymbol{\mu}_i))^2$, and since the matrix's rank is bounded by 2, its nuclear norm will be upper-bounded by $\sqrt{2}$ times its Frobenius norm. As a result of the above inequality, we can write

$$\left| \mathrm{Tr}\Big( \sqrt{\sqrt{C_X} C_Y \sqrt{C_X}} \Big) - \mathrm{Tr}\left( \sqrt{ \sqrt{\sum_{i=1}^{k} \omega_i \phi(\boldsymbol{\mu}_i)\phi(\boldsymbol{\mu}_i)^\top} \Big( \sum_{i=1}^{k} \eta_i \phi(\boldsymbol{\zeta}_i)\phi(\boldsymbol{\zeta}_i)^\top \Big) \sqrt{\sum_{i=1}^{k} \omega_i \phi(\boldsymbol{\mu}_i)\phi(\boldsymbol{\mu}_i)^\top} } \right) \right|$$

$$\leq \sqrt{ \left\| C_Y - \sum_{i=1}^{k} \eta_i \phi(\boldsymbol{\zeta}_i)\phi(\boldsymbol{\zeta}_i)^\top \right\|_* } + \sqrt{ \left\| C_X - \sum_{i=1}^{k} \omega_i \phi(\boldsymbol{\mu}_i)\phi(\boldsymbol{\mu}_i)^\top \right\|_* }$$

$$\leq \sqrt[4]{ \sum_{i=1}^{k} 4\omega_i \big( 1 - \frac{1}{\alpha_i^d} \big) } + \sqrt[4]{ \sum_{i=1}^{k} 4\eta_i \big( 1 - \frac{1}{\alpha_i^d} \big) }$$

$$\leq \sqrt[4]{ \sum_{i=1}^{k} 32(\omega_i + \eta_i)\big( 1 - \frac{1}{\alpha_i^d} \big) }$$

where the last inequality holds as $\sqrt[4]{a} + \sqrt[4]{b} \leq \sqrt[4]{8(a+b)}$ holds for every $a, b \geq 0$, which is a consequence of $\sqrt{a} + \sqrt{b} \leq \sqrt{2(a+b)}$ for every $a, b \geq 0$. The proof is therefore complete.

### A.6 Proof of Theorem 5

Note that according to the definition of kernel covariance matrix we have

$$\mathrm{Tr}\big( C_X C_X^\top \big) = \mathrm{Tr}\left( \int p(\mathbf{x})\phi(\mathbf{x})\phi(\mathbf{x})^\top \mathrm{d}\mathbf{x} \int p(\mathbf{x}')\phi(\mathbf{x}')\phi(\mathbf{x}')^\top \mathrm{d}\mathbf{x}' \right)$$

$$= \operatorname{Tr}\left(\int p(\mathbf{x})p(\mathbf{x}')\phi(\mathbf{x})\phi(\mathbf{x})^\top\phi(\mathbf{x}')\phi(\mathbf{x}')^\top \mathrm{d}\mathbf{x}\mathrm{d}\mathbf{x}'\right)$$

$$= \int p(\mathbf{x})p(\mathbf{x}')\operatorname{Tr}\left(\phi(\mathbf{x})\phi(\mathbf{x})^\top\phi(\mathbf{x}')\phi(\mathbf{x}')^\top\right)\mathrm{d}\mathbf{x}\mathrm{d}\mathbf{x}'$$

$$= \int p(\mathbf{x})p(\mathbf{x}')\operatorname{Tr}\left(\phi(\mathbf{x}')^\top\phi(\mathbf{x})\phi(\mathbf{x})^\top\phi(\mathbf{x}')\right)\mathrm{d}\mathbf{x}\mathrm{d}\mathbf{x}'$$

$$= \int p(\mathbf{x})p(\mathbf{x}')k(\mathbf{x}',\mathbf{x})k(\mathbf{x},\mathbf{x}')\mathrm{d}\mathbf{x}\mathrm{d}\mathbf{x}'$$

$$= \int p(\mathbf{x})p(\mathbf{x}')k(\mathbf{x},\mathbf{x}')^2\mathrm{d}\mathbf{x}\mathrm{d}\mathbf{x}'$$

$$= \mathbb{E}_{X,X'\overset{\text{iid}}{\sim}P_X}\left[k(\mathbf{X},\mathbf{X}')^2\right],$$

where the last line holds since the joint density function of independent $\mathbf{X}, \mathbf{X}'$ will be the product of the marginal density functions. Given the above result, the theorem is a direct consequence of Proposition 1.

### A.7 Proof of Theorem 6

As shown in Theorem 5, we have

$$\exp\left(-\mathrm{RKE}_2(\mathbf{X})\right) = \mathbb{E}_{X,X'\overset{\text{iid}}{\sim}P_X}\left[k^2(\mathbf{X},\mathbf{X}')\right].$$

As a result, we obtain the following:

$$\exp\left(-\widehat{\mathrm{RKE}}_2(\mathbf{X})\right) - \exp\left(-\mathrm{RKE}_2(\mathbf{X})\right) = \frac{1}{n^2}\sum_{i=1}^n\sum_{j=1}^n k(\mathbf{x}_i,\mathbf{x}_j)^2 - \mathbb{E}_{X,X'\overset{\text{iid}}{\sim}P_X}\left[k^2(\mathbf{X},\mathbf{X}')\right]$$

Note that according to the Cauchy-Schwarz inequality, for every $\mathbf{x},\mathbf{y}\in\mathcal{X}$ we have $k(\mathbf{x},\mathbf{y})^2 \leq k(\mathbf{x},\mathbf{x})k(\mathbf{y},\mathbf{y}) = 1$, and therefore for a normalized kernel we will have $0 \leq k(\mathbf{x},\mathbf{y}) \leq 1$ at every $\mathbf{x},\mathbf{y}\in\mathcal{X}$. Therefore, if $\mathbf{z}_1,\ldots,\mathbf{z}_n$ are $n$ IID samples from $P_X$ which are also independent from $\mathbf{x}_1,\ldots,\mathbf{x}_n$, we will have:

$$\left|\frac{1}{n^2}\sum_{i=1}^n k(\mathbf{x}_i,\mathbf{x}_i)^2 - \frac{1}{n^2}\sum_{i=1}^n k(\mathbf{x}_i,\mathbf{z}_i)^2\right| \leq \frac{n}{n^2} - \frac{0}{n^2} = \frac{1}{n}.$$

On the other hand, we know a complete graph of size $n$ can be decomposed to $r = \lceil\frac{n}{2}\rceil$ matchings $m_1,\ldots,m_r$ with $\lfloor\frac{n}{2}\rfloor$ edges. We will have the following identity given these matchings

$$\frac{1}{n^2}\sum_{i\neq j}k(\mathbf{x}_i,\mathbf{x}_j)^2 = \frac{1}{n^2}\sum_{i=1}^r\sum_{t=1}^{\lfloor\frac{n}{2}\rfloor}k(\mathbf{x}_{m_i(t,0)},\mathbf{x}_{m_i(t,1)}).$$

As a result, we have

$$\left|\left[\frac{1}{n^2}\sum_{i=1}^n\sum_{j=1}^n k(\mathbf{x}_i,\mathbf{x}_j)^2\right] - \left[\frac{1}{n^2}\sum_{i=1}^n k(\mathbf{x}_i,\mathbf{z}_i)^2\right] - \left[\frac{1}{n^2}\sum_{i=1}^r\sum_{t=1}^{\lfloor\frac{n}{2}\rfloor}k(\mathbf{x}_{m_i(t,0)},\mathbf{x}_{m_i(t,1)})\right]\right| \leq \frac{1}{n}$$

Now, we note that $\{(\mathbf{x}_i,\mathbf{z}_i)_{i=1}^n\}$ are $n$ independent samples of $(\mathbf{X},\mathbf{X}')$, and an application of the Hoeffding's inequality implies that with probability $1-\delta/n$ we have the following

$$\left|\frac{1}{n}\sum_{i=1}^n k(\mathbf{x}_i,\mathbf{z}_i)^2 - \mathbb{E}_{X,X'\overset{\text{iid}}{\sim}P_X}\left[k^2(\mathbf{X},\mathbf{X}')\right]\right| \leq \sqrt{\frac{2\log(n/\delta)}{n}}.$$

Similarly, every matching $m_t$ provides $\lfloor\frac{n}{2}\rfloor$ independent sample pairs of $(\mathbf{X},\mathbf{X}')$, which implies that with probability $1-\delta/n$

$$\left|\frac{1}{\lfloor n/2\rfloor}\sum_{i=1}^{\lfloor n/2\rfloor}k(\mathbf{x}_{m_t(i,0)},\mathbf{x}_{m_t(i,1)})^2 - \mathbb{E}_{X,X'\overset{\text{iid}}{\sim}P_X}\left[k^2(\mathbf{X},\mathbf{X}')\right]\right| \leq \sqrt{\frac{4\log(n/\delta)}{n}}.$$

Given the above bounds, an application of the union bound shows that with probability at least $1 - n \times \frac{\delta}{n} = 1 - \delta$, we will have the following

$$\left| \mathbb{E}_{X,X' \stackrel{\text{iid}}{\sim} P_X} \left[ k^2(\mathbf{X}, \mathbf{X}') \right] - \left[ \frac{1}{n^2} \sum_{i=1}^{n} k(\mathbf{x}_i, \mathbf{z}_i)^2 \right] - \left[ \frac{1}{n^2} \sum_{i=1}^{r} \sum_{t=1}^{\left[\frac{n}{2}\right]} k(\mathbf{x}_{e_i(t,0)}, \mathbf{x}_{e_i(t,1)}) \right] \right| \leq O\left( \sqrt{\frac{\log(n/\delta)}{n}} \right),$$

which can be combined with the mentioned upper-bound to show that with probability at least $1 - \delta$ the following holds

$$\left| \frac{1}{n^2} \sum_{i=1}^{n} \sum_{j=1}^{n} k(\mathbf{x}_i, \mathbf{x}_j)^2 - \mathbb{E}_{X,X' \stackrel{\text{iid}}{\sim} P_X} \left[ k^2(\mathbf{X}, \mathbf{X}') \right] \right| \leq O\left( \sqrt{\frac{\log(n/\delta)}{n}} + \frac{1}{n} \right)$$

$$= O\left( \sqrt{\frac{\log(n/\delta)}{n}} \right).$$

Therefore, the proof is complete.

### A.8 Proof of Theorem 7

Note that according to our definition, we will have

$$K_{XY} = \Phi_X \Phi_Y^\top.$$

We consider the SVD decomposition of matrices $\Phi_X = U_X S_X V_X^\top$ and $\Phi_Y = U_Y S_Y V_Y^\top$ where $U_X, U_Y$ are unitary matrices in $\mathbb{R}^{n \times n}$, $V_X, V_Y$ are unitary matrices in $\mathbb{R}^{d \times d}$, and $S_X, S_Y$ are semi-diagonal matrices in $\mathbb{R}^{n \times d}$. Then, we will have

$$K_{XY} = U_X S_X V_X^\top V_Y S_Y U_Y^\top.$$

Also, we can obtain that

$$C_X = \Phi_X^\top \Phi_X = V_X S_X^\top S_X V_X^\top, \quad C_Y = \Phi_Y^\top \Phi_Y = V_Y S_Y^\top S_Y V_Y^\top.$$

As a result, we will have the following

$$\sqrt{C_X} \sqrt{C_Y} = V_X \sqrt{S_X^\top S_X} V_X^\top V_Y \sqrt{S_Y^\top S_Y} V_Y^\top$$

However, since $S_X, S_Y$ are semi-diagonal and $U_X, U_Y$ are unitary matrices, we will have the same set of non-zero singular-values for the following two matrices

$$\text{singular.values}\left( V_X \sqrt{S_X^\top S_X} V_X^\top V_Y \sqrt{S_Y^\top S_Y} V_Y^\top \right) = \text{singular.values}\left( U_X S_X V_X^\top V_Y S_Y U_Y^\top \right)$$

Therefore, $K_{XY}$ shares the same singular values with $\sqrt{C_X} \sqrt{C_Y}$. Therefore, we will have

$$\text{Tr}\left( \sqrt{\sqrt{C_X} C_Y \sqrt{C_X}} \right) = \sum_{i=1}^{d} s_i\left( \sqrt{C_X} \sqrt{C_Y} \right) = \sum_{i=1}^{d} s_i(K_{XY}) = \|K_{XY}\|_*,$$

which due to the definition of order-$\frac{1}{2}$ RRKE score completes the proof.

### A.9 Additional Experimental Results

#### A.9.1 Effect of bandwidth on RKE

We show the effect of different bandwidths on CIFAR10, Tiny-ImageNet, and ImageNet datasets in Figure 4. This plot indicates that the ranking of the models remains consistent for different bandwidth parameters in the range $\sigma \in [0.1, 0.5]$. It is important to note that for bandwidth values $\sigma > 0.5$, the Gaussian kernel assigns near-zero values to almost every pair of input samples, and therefore all the RKE mode count values are close to 1. On the other hand, for smaller $\sigma \approx 0$ bandwidth values, every data point would be counted as a separate mode (high sensitivity to between samples distances). We also experimented the effect of different bandwidths on StyleGAN3 with different truncation factors in Figure 5.

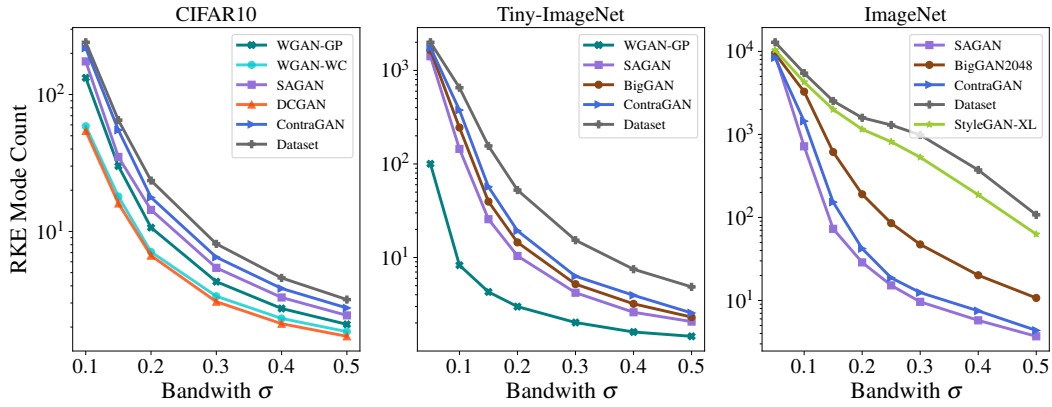

Figure 4: Effect of the bandwidth $\sigma$ on the numerical evaluation

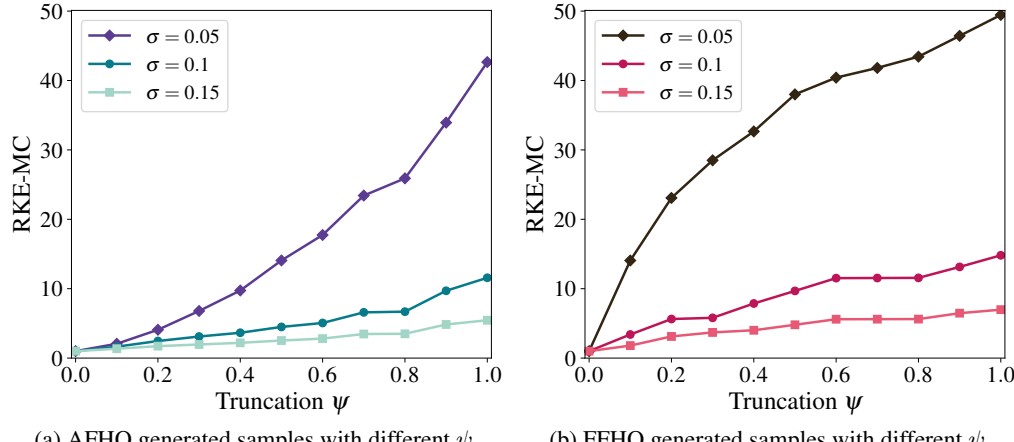

(a) AFHQ generated samples with different $\psi$.  (b) FFHQ generated samples with different $\psi$.

Figure 5: Effect of the bandwidth $\sigma$ on StyleGAN3 with different truncation factor on AFHQ and FFHQ dataset.

### A.9.2  Can existing metrics count the number of modes?

In our experiment on Gaussian distributions, we observed that existing metrics like Recall and Coverage are unable to quantify the number of modes. As shown in Figure 6, we have two generated data, one with a single mode and the other with two Gaussian modes. Recall and Coverage yielded the same results for both datasets. However, RKE-MC demonstrated the capability to count the number of modes in this experiment accurately.

### A.9.3  Different orders of Rényi entropy scores in applications of the RKE evaluation

We evaluated the matrix-based Rényi entropy score of different orders and summarized the numerical results in Figure 7. As shown in this figure, order-2 Renyi entropy can successfully distinguish the diversity performance of BigGAN-2048 and SAGAN.

### A.9.4  Comparison between our proposed algorithms for computing the RKE score and other algorithms

In our numerical evaluation of the RKE score, we focused on order-2 matrix-based Renyi entropy which reduces to the Frobenius norm of the kernel matrix. This algorithm will require computation for samples of dimension. In addition, Theorem 5 implies a randomized algorithm estimating the expected value using empirical samples which requires computation for pairs of fresh empirical samples. On the other hand, Dong et al. [25]'s computation method applies to a general order-$\alpha$ matrix-based Renyi entropy.

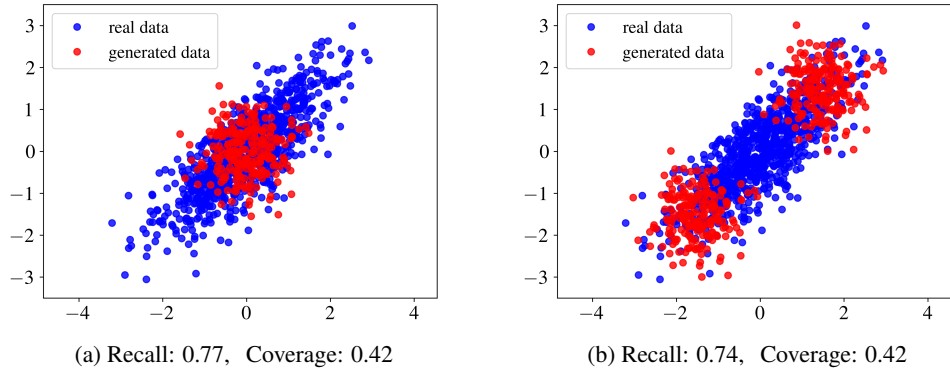

(a) Recall: 0.77,   Coverage: 0.42          (b) Recall: 0.74,   Coverage: 0.42

Figure 6: Recall and Coverage can not count the number of modes.

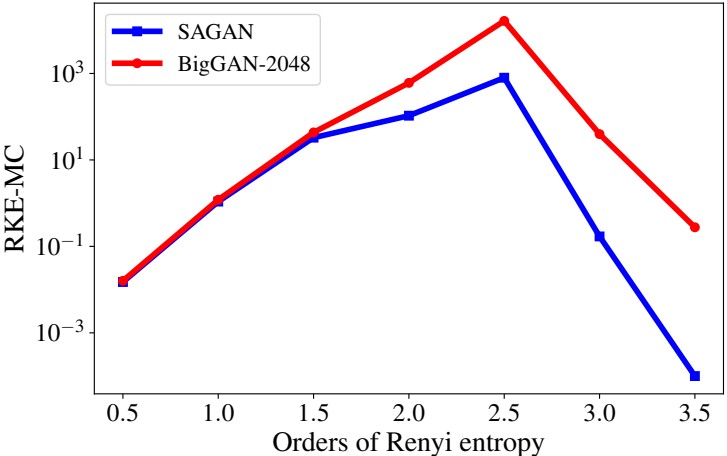

Figure 7: Effect of different Renyi entropy orders in RKE-MC in application to ImageNet data

The table 3 shows the time (in seconds) taken by the three algorithms in performing the computation of order-2 Renyi entropy. The results in the table indicate that Dong et al.'s method and the Frobenius norm-based approach result in similar time complexity, while the randomized algorithm based on empirical expected value can significantly reduce the computation time.

### A.9.5   RKE-MC During Training

We trained the ContraGAN and SNGAN on CIFAR-10 data and recorded the evaluation scores every 2,000 generator iterations. As shown in Figure 8, RKE increased during the training. In Figure 9, we can see the diversity of generated samples for 10 classes of CIFAR10 during the training of ContraGAN.

Table 3: Comparison between algorithms for computing the RKE score

| Algorithms | 1000 samples | 2000 samples | 3000 samples |
|---|---|---|---|
| Frobenius norm (Ours) | 9.12 | 36.35 | 81.85 |
| Empirical expected value (Ours) | 2.19 | 3.25 | 5.10 |
| Dong et al. (Hutch++ based) | 8.97 | 35.9 | 82.08 |

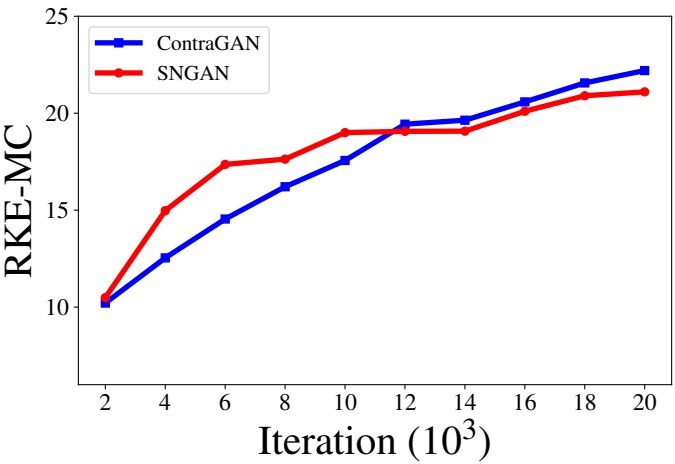

Figure 8: RKE Mode Count during training of ContraGAN and SNGAN on CIFAR10 dataset.

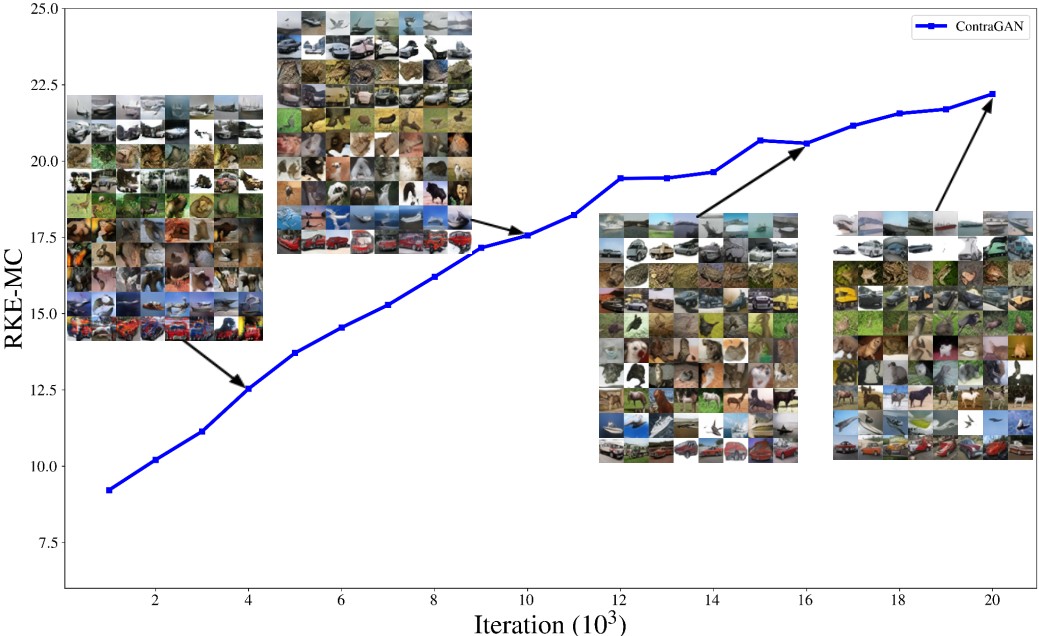

Figure 9: RKE Mode Count during training of Contra-GAN on CIFAR10 dataset with generated samples from 10 classes.

Table 4: Absolute evaluation for three image datasets. IS-diversity & IS-quality report $10^3 \exp(H(Y))$ and $10 \exp(-H(Y|X))$ and RKE-MC (Mode Count) denotes $\exp(\text{RKE})$.

| | Method | IS ↑ | Separated Inception Score IS-diversity ↑ | IS-quality ↑ | FID ↓ | KID ↓ | **RKE-MC** ↑ |
|---|---|---|---|---|---|---|---|
| **CIFAR-10** | Dataset | 11.57 | 39.87 | 29.01 | - | - | 39.58 |
| | NVAE | 5.85 | 25.05 | 23.36 | 51.67 | 0.0502 | 17.65 |
| | VDVAE | 10.51 | 38.40 | 27.37 | 37.51 | 0.0247 | 32.49 |
| | DCGAN | 5.75 | 16.41 | 34.89 | 54.30 | 0.0536 | 10.19 |
| | WGAN-WC | 2.59 | 35.25 | 7.37 | 157.26 | 0.0914 | 10.64 |
| | WGAN-GP | 7.51 | 25.82 | 28.85 | 21.66 | 0.0106 | 19.07 |
| | SAGAN | 8.62 | 28.33 | 30.13 | 10.17 | 0.0077 | 24.46 |
| | SNGAN | 8.81 | 30.08 | 29.31 | 9.23 | 0.0051 | 25.83 |
| | ContraGAN | 9.69 | 35.70 | 27.14 | 4.02 | 0.0023 | 29.80 |
| **Tiny-ImageNet** | Dataset | 33.99 | 60.56 | 56.12 | - | - | 155.86 |
| | SAGAN | 8.21 | 18.34 | 44.75 | 46.98 | 0.0531 | 25.68 |
| | SNGAN | 8.12 | 18.74 | 43.34 | 48.96 | 0.0567 | 27.18 |
| | BigGAN | 11.57 | 25.08 | 46.17 | 27.34 | 0.0262 | 39.61 |
| | ContraGAN | 13.79 | 28.96 | 47.62 | 21.36 | 0.0175 | 56.94 |
| **ImageNet** | Dataset | 357.35 | 370.78 | 96.37 | - | - | 1823.52 |
| | SAGAN-256 | 29.67 | 35.66 | 83.22 | 44.66 | 0.0372 | 105.57 |
| | SNGAN-256 | 31.92 | 35.70 | 89.41 | 35.75 | 0.0391 | 115.62 |
| | ContraGAN-256 | 24.91 | 30.21 | 82.46 | 34.79 | 0.0403 | 152.89 |
| | BigGAN-256 | 28.33 | 31.68 | 89.43 | 33.48 | 0.0440 | 106.07 |
| | ReACGAN-256 | 52.53 | 62.24 | 75.05 | 15.65 | 0.0382 | 119.76 |
| | BigGAN-2048 | 96.42 | 104.24 | 92.49 | 0.89 | 0.0038 | 606.18 |
| | StyleGAN-XL | 204.73 | 292.50 | 69.99 | 1.94 | 0.0035 | 1375.17 |
| | LDM-4-G | 242.62 | 252.43 | 94.65 | 3.60 | 0.0036 | 1321.24 |
| | ADM-G | 188.70 | 216.23 | 92.34 | 3.86 | 0.0036 | 1407.75 |

### A.9.6 Datasets Results

We have divided the scores into absolute (Table 4) and relative scores (Table 5) where absolutes scores reports only based on the input samples but relative scores are based on the dataset and the generated samples.

### A.9.7 The effect of standard deviation in RKE

In this experiment, we investigate the impact of standard deviation over $\sigma$. The real distribution is two Gaussian mixtures with $(-5, 0), (5, 0)$ as their centers with varying component std. The data and the first 10 eigenvalues of the kernel are shown in Figure 10. RKEs with different hyperparameter $p$'s are shown in Table 6.

Table 5: Relative evaluation scores for CIFAR-10. A lower RRKE implies higher joint diversity.

| | Method | Precision ↑ | Recall ↑ | Density ↑ | Coverage ↑ | **RRKE** ↓ |
|---|---|---|---|---|---|---|
| CIFAR-10 | NVAE | 0.36 | 0.50 | 0.28 | 0.60 | 2.01 |
| | VDVAE | 0.34 | 0.78 | 0.23 | 0.21 | 1.81 |
| | DCGAN | 0.59 | 0.25 | 0.49 | 0.23 | 0.98 |
| | WGAN-WC | 0.36 | 0.00 | 0.18 | 0.03 | 2.09 |
| | WGAN-GP | 0.62 | 0.56 | 0.57 | 0.51 | 0.74 |
| | SAGAN | 0.68 | 0.62 | 0.73 | 0.73 | 0.65 |
| | SNGAN | 0.70 | 0.62 | 0.77 | 0.74 | 0.62 |
| | ContraGAN | 0.75 | 0.62 | 0.99 | 0.86 | 0.52 |
| Tiny-ImageNet | SAGAN | 0.55 | 0.49 | 0.44 | 0.27 | 1.42 |
| | SNGAN | 0.55 | 0.46 | 0.40 | 0.26 | 1.46 |
| | BigGAN | 0.60 | 0.58 | 0.53 | 0.43 | 1.23 |
| | ContraGAN | 0.62 | 0.54 | 0.54 | 0.45 | 1.26 |
| ImageNet | SAGAN | 0.57 | 0.58 | 0.42 | 0.35 | 2.34 |
| | SNGAN | 0.54 | 0.64 | 0.41 | 0.38 | 2.22 |
| | ContraGAN | 0.67 | 0.51 | 0.64 | 0.33 | 2.54 |
| | BigGAN256 | 0.58 | 0.61 | 0.49 | 0.37 | 2.28 |
| | ReACGAN-256 | 0.74 | 0.42 | 0.79 | 0.73 | 2.20 |
| | BigGAN2048 | 0.71 | 0.58 | 0.80 | 0.65 | 1.83 |
| | StyleGAN-XL | 0.77 | 0.61 | 0.67 | 0.81 | 1.50 |
| | LDM-4-G | 0.86 | 0.60 | 0.69 | 0.78 | 1.56 |
| | ADM-G | 0.82 | 0.64 | 0.66 | 0.82 | 1.47 |

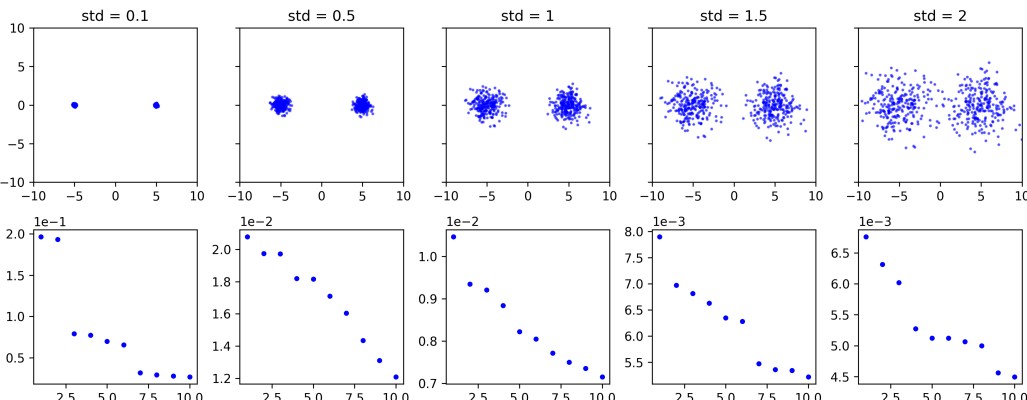

Figure 10: 500 samples from 2D Gaussian distribution with (-5, 0) and (5, 0) as their centers and their eigenvalue in the second row.

Table 6: RKE-MC result with different hyperparameter $\sigma$ for Figure 10 samples.

| $\sigma$ | std = 0.1 | std = 0.5 | std = 1 | std = 1.5 | std = 2 |
|---|---|---|---|---|---|
| 0.1 | 9.69 | 139.95 | 295.92 | 380.01 | 423.77 |
| 0.5 | 2.31 | 9.69 | 31.26 | 63.07 | 100.57 |
| 1 | 2.07 | 3.94 | 9.69 | 18.95 | 31.19 |
| 2 | **2.01** | 2.48 | 3.94 | 6.35 | 9.64 |
| 5 | 1.96 | **2.03** | **2.24** | **2.56** | **2.99** |
| 10 | 1.46 | 1.47 | 1.50 | 1.55 | 1.62 |