# OpenReview forum: "An Information-Theoretic Evaluation of Generative Models in Learning Multi-modal Distributions"
_NeurIPS.cc/2023/Conference — NeurIPS 2023 poster_

### Official Review · Reviewer_3kP8 · 2023-07-06

**Soundness:** 3 good
**Presentation:** 3 good
**Contribution:** 3 good
**Rating:** 6
**Confidence:** 4

**Summary:**

This paper investigates an information theoretic approach in evaluating the diversity of generative models by using Rényi kernel entropy (RKE) and relative Rényi kernel entropy (RRKE). The RKE measures the absolute diversity of a multi-modal distribution, while RRKE measures the relative diversity given the data distribution. They theoretically approve computational efficiency of RKE and RRKE estimators with convergence guarantee. The proposed evaluation metric is applied to various image generative models on CIFAR-10, tiny-ImageNet, and ImageNet-1K datasets, showing its effectiveness.

**Strengths:**

Evaluating the diversity of generative model is an important problem, yet many works resort to the FID or Inception Score due to its simplicity. The proposed method provides an alternative approach in metric for generative models with information theoretic principle. The proposed theorems are clearly written and the usage of Rényi divergence is novel. Also, it is nice to see the sample complexity (e.g., how many samples do we need to compute the metric).

**Weaknesses:**

The usage of RKE and RRKE depends on various hyper parameters, such as the choice of kernel, where the author choose RBF kernel as default, thus the bandwidth hyperparameter must be chosen with validation step. Also, different order of Rényi entropies might result in different results, where they focus on order 2 and 1/2 for both RKE and RRKE. Generalizing to various orders or studying the effect of an order of Rényi entropy could enrich the paper.

In practical side, while the RKE and RRKE aligns with other diversity metrics, the paper does not compare the superiority between metrics. For example, how RKE and RRKE is more aligned with human judgement? If there is such human ranking, comparing rank-correlation (e.g., Spearman’s rho or Kendall’s delta) can better show the effectiveness of RKE and RRKE.



**Questions:**

1. Since there are various open-source multi-modal generative models (e.g., text-to-image diffusion models). How one can leverage RKE or RRKE in such cases? To be specific, evaluating the text-to-image models is a non-trivial problem (as the context made from text is infinitely large compared to class-conditional one). Do you have any intuition on how to better diversity of text-to-image models with RKE or RRKE?
2. Can RRKE be used throughout training to improve diversity? Instead of using it as evaluation, is it possible to use Relative Rényi kernel entropy as a regularizer to promote diversity in learning generative models?


**Limitations:**

Some limitations of the proposed evaluation method could be useful for the readers, for example, see the weakness part.

---

> ### Author Rebuttal · Authors · 2023-08-10
>
> We thank Reviewer 3kP8 for his/her time and constructive suggestions and feedback on our work. In the following, we respond to the comments and questions raised in the review:
>
> **1- Selecting the order of Renyi entropy scores in applications of the RKE evaluation**
>
> **Re:** In this work, we have focused on the order-2 RKE score that can be supported by a sample complexity bound (Theorem 6) as well as a fast computation method via a randomized approach for estimating the expected value in Theorem 5’s formulation. To address the reviewer’s comment, we evaluated the matrix-based Renyi entropy score of different orders and summarized the numerical results in Figure 1 of the submitted rebuttal PDF. As shown in this figure, order-2 Renyi entropy can successfully distinguish the diversity performance of BigGAN-2048 and SAGAN. An interesting future direction is to theoretically analyze the effect of changing the order of Renyi entropy in the diversity evaluation task.
>
> **2-Tuning the kernel bandwidth hyperparameter of the RKE/RRKE scores**
>
> **Re:** We would like to refer the reviewer to Figure 4 in the supplementary PDF which shows the effect of different bandwidth hyperparameters in the resulting RKE/RRKE scores. This figure indicates a consistent ranking of our attempted generative models across different bandwidth values.
>
>
> **3- RKE/RRKE vs. other diversity evaluation scores**
>
> **Re:** As discussed in the introduction, our primary motivation for proposing the RKE/RRKE-based evaluation of generative models is to develop an information theoretic approach to the assessment problem. While entropy measures are standard diversity scores in information theory, the challenging entropy estimation task in high-dimensional computer vision settings prevents their application to the evaluation of state-of-the-art generative models. In this work, we show that quantum information measures can address this challenge by estimating the entropy of major modes in an underlying multimodal distribution.
>
> Furthermore, we show that the proposed scores can be efficiently estimated from a limited number of training data which is the key to achieve lower statistical and computational costs in the evaluation process. Our truncation-based numerical results in the paper’s Figure 2 confirm the success of the proposed score in quantifying the diversity of generated samples. Also, in Figure 3 we demonstrate the efficient estimation procedure of the RKE score vs. standard diversity evaluation metrics. Finally, we note that performing a theoretical comparison between different diversity assessment criteria will be an interesting future direction to our work.
>
> **4- Diversity evaluation of text-to-image generative models**
>
> **Re:** Based on the reviewer’s recommendation, we used the proposed RKE and RRKE scores to evaluate text-to-image generative models which is similar to computing FID-30k for these models. In our experiments, we evaluated the state-of-the-art text-to-image generative model GigaGAN[1] and BK-SDM[2] on the COCO dataset. The numerical results of the experiments can be found in the rebuttal PDF’s Table 2, showing that the GigaGAN model achieves a higher diversity score than BK-SDM.
>
>
> [1] Kang, M., Zhu, J.Y., Zhang, R., Park, J., Shechtman, E., Paris, S., & Park, T. Scaling up GANs for Text-to-Image Synthesis. In Proceedings of the IEEE Conference on Computer Vision and Pattern Recognition (CVPR), 2023,\
> [2] Bo-Kyeong Kim, and Hyoung-Kyu Song, and Thibault Castells, and Shinkook Choi, “BK-SDM: Architecturally Compressed Stable Diffusion for Efficient Text-to-Image Generation”,  ICML 2023.
>
> **5- Applying the RRKE score to regularize the training of generative models**
>
> **Re:** Since state-of-the-art generative models are typically trained by first-order optimization algorithms, the application of RRKE to regularize their training will be challenging. We note that the empirical RRKE is a non-differentiable map of the generative model due to the presence of nuclear norm in its formulation (shown in Theorem 7). Finding differentiable proxies of the RRKE score to regularize the training of generative models is definitely an interesting future direction, which we will discuss in the conclusion section.

---

> > ### Comment · Reviewer_3kP8 · 2023-08-17
> > **Response to the rebuttal**
> >
> > I appreciate the authors' rebuttal with their efforts on additional experiments despite the short term of preparation. I am satisfied with the authors' response, and I will maintain my original score.

---

> > > ### Author Response · Authors · 2023-08-20
> > > **Thank you for your feedback**
> > >
> > > We thank Reviewer 3kP8 for his/her time and feedback on our responses. We will revise the draft based on the reviewer's comments and suggestions and include the numerical results on VAEs and text-to-image generative models in the text.

---

### Official Review · Reviewer_5vQM · 2023-07-07

**Soundness:** 4 excellent
**Presentation:** 4 excellent
**Contribution:** 3 good
**Rating:** 8
**Confidence:** 4

**Summary:**

The authors propose Renyi Kernel Entropy (RKE) as a measure of diversity for generative models. A relative diversity variant, relative Renyi kernel entropy (RRKE) is also defined. Estimation error bounds are derived and the measure is applied to both toy data, where it behaves as expected, and to state of the art image generative models.

**Strengths:**

* The proposed methodology is novel to the problem of estimating the diversity of generative models, a problem of interest to the NeurIPS audience.
* The scores are clearly introduced and well-motivated by their ability to count the number of well-separated modes in a Gaussian mixture model, which is confirmed by the results in Figure 1.
* The estimation error bounds are a great contribution.
* Figure 3 shows great empirical convergence.

Overall, the paper tackles a well-defined problem and offers an attractive solution in a detailed, convincing way.

**Weaknesses:**

* In the manuscript, it is unclear how matrix-based Renyi entropy is used in quantum information theory and how that relates to a notion of sample diversity.

**Questions:**

Suggestions:
* The abstract states "we show that the RKE score can output the number of modes of a mixture of sub-Gaussian components." It should be clarified that this is an estimate that refers to well-separated modes.

* Figure 1 should display RKE-MC (mode count) as opposed to REK to make it consistent with Table 1.

l223: It seems this should say one over square root n convergence.

**Limitations:**

Yes, although see the first suggestion under "Questions."

---

> ### Author Rebuttal · Authors · 2023-08-10
>
> We thank Reviewer 5vQM for his/her time and constructive suggestions and feedback on our work. In the following, we respond to the comments and questions in the review:
>
> **1- Discussion on how matrix-based Renyi entropy is used in quantum information theory**
>
> **Re:** The application of quantum information measures is motivated through considering different states in the generated data, which could be interpreted as the modes of a multimodal distribution that are separable by a distance $\gg\sigma$ when we use a Gaussian kernel with bandwidth $\sigma$ to define similarities between samples. In the revision, we will further explain the motivation behind applying quantum information measures and how we attempt to theoretically formalize the motivation by analyzing mixtures of well-separable sub-Gaussians.
>
> **2- “It should be clarified that this is an estimate that refers to well-separated modes.”**
>
> **Re:** As recommended by the reviewer, we will clarify the separability condition of the modes in the abstract and introduction.
>
> **3- RKE Mode Count in Figure 1 and Typo in Line 223**
>
> **Re:** We thank the reviewer for pointing out the typos in Figure 1 and Line 223. Figure 1 already reports RKE mode count ($\exp(\mathrm{RKE})$). We will correct the typos in the revision.

---

> > ### Comment · Reviewer_5vQM · 2023-08-19
> >
> > Thank you to the authors for your response. I think the additional data on VAEs and text-to-image models is useful for putting the scores into context. Reading the other reviews and responses, I remain positive about this manuscript.

---

> > > ### Author Response · Authors · 2023-08-20
> > > **Thank you for your feedback**
> > >
> > > We thank Reviewer 5vQM for his/her time and feedback on our response. We will revise the draft based on the reviewer’s comments and include the numerical results on VAEs and text-to-image generative models as discussed in the rebuttal.

---

### Official Review · Reviewer_YzmH · 2023-07-12

**Soundness:** 3 good
**Presentation:** 2 fair
**Contribution:** 3 good
**Rating:** 6
**Confidence:** 4

**Summary:**

This paper introduces Renyi Entropy to measure the number of modes in generated samples. To this end, two metrics called empirical Renyi Kernel Entropy (RKE) and Relative Renyi Kernel Entropy (RRKE) are defined to estimate the Renyi Entropy of the eigenvalues of the data's kernel function. Some theoretical analyses are provided to guarantee the convergence of the proposed metrics. Experiments on synthesized data and benchmarks show that the proposed metrics can produce reasonable measurements.

**Strengths:**

This paper focuses on an important topic, the measurement of generative models, and proposes a seemly promising idea, i.e. using Renyi Entropy to estimate the diversity of generated samples. The theoretical analysis makes the proposed matrics convincing and solid. The experimental results given by the proposed metrics seem reasonable.


**Weaknesses:**

My biggest concern is the applicability of the proposed metrics to more generative models, as this paper mainly reports results on GANs (as well as two diffusion models on ImageNet), but no results on VAEs (e.g. NVAE). It is well known that the samples generated by VAEs are more diverse than GANs but has some flaws like blurriness. Whether the proposed metrics can deal with VAEs remains unknown.

Another weakness of this paper is the absence of an explanation of concepts, hence the paper is not self-contained enough and somewhat hard to follow. For example, what is the meaning of interpreting the eigenvalues of the kernel matrix as probability? Why apply Renyi Entropy to these eigenvalues? What is relative diversity? Readers are difficult to understand the motivation and soundness of this paper without being told these concepts.

Moreover, I suggest the authors discuss the relations and differences between RKE/ RRKE and other metrics (IS, FID, etc.). This will help readers to understand the benefits and position of the proposed metrics, as well as provide the necessity for proposing new metrics in this paper.

**Questions:**

See questions in the weakness part above.

**Limitations:**

The authors have adequately addressed the limitations.

---

> ### Author Rebuttal · Authors · 2023-08-10
>
> We thank Reviewer YzmH for his/her time and constructive feedback and suggestions. In the following, we respond to the comments and questions raised in the review:
>
> **1- Application of the proposed scores to VAEs**
>
> **Re:** Based on the reviewer’s recommendation, we have estimated the proposed RKE and RRKE scores as well as the baseline scores for the suggested NVAE [1] and VDVAE [2] generative models. Consistent with Reviewer YzmH’s intuition, we observed that the RKE score of the trained VDVAE significantly improves upon the RKE scores of tested GAN generative models. On the other hand, the RRKE of the VAE models do not improve upon the GAN models, suggesting that the VAE models could produce more diverse results that have lower intersection with the modes of the data distribution. The evaluation scores for the VAE and GAN models can be found in Table 1 of the submitted rebuttal PDF, which we will include in the revised paper.
>
> [1] Arash Vahdat and Jan Kautz. NVAE: A deep hierarchical variational autoencoder. In Neural Information Processing Systems (NeurIPS), 2020 \
> [2] Rewon Child. Very deep VAEs generalize autoregressive models and can outperform them on images. In International Conference on Learning Representations (ICLR), 2021.
>
>
> **2- Application of Renyi Entropy to the eigenvalues of kernel matrix**
>
> **Re:** We would like to note that the main contribution of our paper is to provide an operational interpretation of the matrix-based Renyi entropy for the evaluation of generative models. As suggested by Theorems 1-4, our theoretical interpretation reveals that for a multimodal distribution with well-separable sub-Gaussian components, the eigenvalues of the kernel matrix provide an estimation of the mode frequencies and the proposed RKE and RRKE scores will approximately be the standard (non-matrix) Renyi entropy for the distribution over the modes. We believe this score will result in a useful evaluation of standard generative models due to the presence of multiple modes in standard distributions of image and text data.
>
> **3- “What is relative diversity? “**
>
> **Re:** In the paper, we use the term “absolute diversity” to refer to the frequency of modes in the generative model, which in the special case of uniformly-distributed samples across modes means how many modes exist in the generated samples. On the other hand, we use the term “relative diversity” to represent what portion of modes existing in the training data have been captured by the generative model. We will make this point clear in the text.
>
> **4- “Differences between RKE/ RRKE and other metrics (IS, FID, etc.)”**
>
> **Re:** The main difference between the proposed RKE and the Inception score is the dynamic selection of modes used in the RKE score compared to the static modes used in the Inception score. Based on the definition of Inception score, the modes are statically chosen to be the 1000 labels of the ImageNet dataset. On the other hand, the RKE score allows a dynamic identification of the modes after observing the generated data, which will be more flexible in application to non-ImageNet data, e,g, CIFAR10, AFHQ, and FFHQ datasets discussed in the paper. We will include this discussion in the text.

---

> > ### Comment · Reviewer_YzmH · 2023-08-18
> >
> > Thanks to the authors' reply, which resolves my concerns. I will raise my score from 5 to 6.

---

> > > ### Author Response · Authors · 2023-08-20
> > > **Thank you for your feedback**
> > >
> > > We thank Reviewer YzmH for his/her feedback on our response. As discussed in the rebuttal, we will revise the draft based on the reviewer's comments and include the numerical results on VAEs and text-to-image generative models in the text.

---

### Official Review · Reviewer_uN6a · 2023-07-12

**Soundness:** 2 fair
**Presentation:** 2 fair
**Contribution:** 2 fair
**Rating:** 5
**Confidence:** 4

**Summary:**

This paper uses matrix-based Renyi entropy and relative entropy to evaluate the diversity of generative models; in which the former measures absolute diversity (as it correlates with the number of modes in a mixture distribution) and the latter measures relative diversity (as it is in the same expression with the quantum Renyi relative entropy).  Authors use their two measures to evaluate the generative quality of a series of GAN and diffusion model architectures; and make some observations.

**Strengths:**

1. The general idea to use entropy-based measures to evaluate diversity of generative models makes sense to me.
2. It is good that authors demonstrated closed-form expressions for both measures in case of mixture Gaussians or sub-Gaussians.

**Weaknesses:**

1. I found the definition of the empirical Renyi kernel entropy (RKE) which relies on a kernel Gram matrix K is exactly the same to the
definition of matrix-based Renyi's alpha-order entropy functional in [Sanchez Giraldo et al., 2014], which also has fast implementation in [Dong et al., 2023].

In this sense, it is improper to use the expressions like "we define ...". etc.

[Sanchez Giraldo et al., 2014] Giraldo, Luis Gonzalo Sanchez, Murali Rao, and Jose C. Principe. "Measures of entropy from data using infinitely divisible kernels." IEEE Transactions on Information Theory 61.1 (2014): 535-548.

[Dong et al., 2023] Dong, Yuxin, et al. "Optimal Randomized Approximations for Matrix-based Rényi's Entropy." IEEE Transactions on Information Theory (2023).

2. Some claims are not well explained. For example, according to Theorems 1 and 2, the RKE correlates with the number of modes k. However, it does not reflect that RKE is propositional to k? or why RKE can be precisely used to quantify the value of k? Similarly, why RRKE can be used to quantify the number of common modes according to Theorem 4.

**Questions:**

1. Can you explain the difference with respect to definition in [Sanchez Giraldo et al., 2014]? How is your efficient computation compared to [Dong et al., 2023]?
2. See the second point of weaknesses
3. In Theorem 4, is it necessary that two distributions have the same number of modes?
4. Can you further explain the computational complexity of your measures compared to previous popular scores?

**Limitations:**

Authors did not discuss potential limitations and negative societal impacts.

---

> ### Author Rebuttal · Authors · 2023-08-10
>
> We thank Reviewer uN6a for his/her time and feedback. The following is our response to the reviewer’s comments and questions:
>
> **1-  Our work’s contributions and the mentioned references**
>
> **Re:** We would like to clarify that we do not claim we are the first to define the matrix-based Renyi entropy (general or kernel-based) which is a well-known tool in quantum information theory. We believe our main contribution is to apply the matrix-based Renyi entropy for the evaluation of generative models and further to provide theoretical interpretation for the Renyi entropy evaluation as a mode-based diversity score for multi-modal distributions. To the best of our knowledge, these contributions are novel and have not been discussed in previous works.
>
> Regarding the discussed papers in the review, we thank the reviewer for pointing out the reference [Sanchez Giraldo et al., 2014]. We note that the only common result between our work and [Sanchez Giraldo et al., 2014] is Corollary 1 on the analytical derivation of the empirical RKE score which we will clarify in the revision. However, we would like to highlight that all the other theoretical results in Theorems 1-7 are independent from [Sanchez Giraldo et al., 2014]’s findings, which includes
>
> **1-** Theorems 1-4 on the mode-based diversity interpretation of RKE and RRKE for mixtures of Gaussians and sub-Gaussians,\
> **2-** Theorem 6 on the sample complexity guarantee for the RKE score,\
> **3-** Theorem 7 on the nuclear-norm-based computation of empirical RRKE score,
>
> We sincerely hope that Reviewer uN6a will consider the above contributions and also the shown application of the results to the evaluation of generative models in his/her assessment of our work. We will be happy to further discuss the reviewer’s concern on the novelty of our theoretical and numerical results.
>
> **2- Comparison between our proposed algorithms for computing the RKE score and [Dong et al., 2023]’s method**
>
> **Re:** In our numerical evaluation of the RKE score, we focused mostly on order-2 matrix-based Renyi entropy which reduces to the Frobenius norm of the kernel matrix as shown in Corollary 1. This algorithm will require $O(n^2 d)$ computation for $n$ samples of dimension $d$. In addition, Theorem 5 implies a randomized algorithm estimating the expected value using empirical samples which requires $O(nd)$ computation for $n$ pair of fresh empirical samples. On the other hand, [Dong et al., 2023]’s computation method applies to a general order-$\alpha$ matrix-based Renyi entropy.
>
> The following table shows the time (in seconds) taken by the three algorithms in performing the computation of order-2 Renyi entropy. The results in the table indicate that [Dong et al., 2023]’s method and the Frobenius norm-based approach result in similar time complexity, while the randomized algorithm based on empirical expected value can significantly reduce the computation time.
>
> | Algorithms | 1000 samples | 2000 samples | 3000 samples |
> |:--------------:|:---------:|:---------:|:---------:|
> Frobenius norm (Ours) | 9.12 | 36.35 | 81.85 |
> Empirical expected value (Ours) | 2.19 | 3.25 | 5.10 |
> Dong et al. (Hutch++ based) | 8.97 | 35.9 | 82.08 |
>
> **3- “why can RKE be used to quantify the value of k?”**
>
> **Re:** The reduction of RKE to the number of modes $k$ depends on the uniformity of samples’ distribution across the $k$ modes. Under a uniform distribution of samples across modes ($\omega_i =\frac{1}{k}$ in Theorem 1), the approximate RKE score in Line 174 for well-separable modes will be $$-\log\Bigl(\sum_{i=1}^k \omega_i^2 \Bigr) = -\log\Bigl(k\times \frac{1}{k^2}\Bigr) = \log(k)$$ which is the entropy of a uniform $k$-ary distribution.  In contrast, under a non-uniform distribution of samples across the $k$ modes, the raw number of modes could become a suboptimal measure of diversity because some of the modes may have little frequency. In that case, our results suggest that the RKE score will approximately reduce to the order-$2$ Renyi entropy of the distribution across the $k$ modes, i.e. $-\log\bigl(\sum_{i=1}^k \omega_i^2\bigr)$. This reduction generalizes the mode-based diversity to non-uniformly distributed samples across modes. We will explain the connection more clearly in the revised text.
>
> **4-  Theorem 4 and equal number of modes for distribution $P,Q$**
>
> **Re:** Theorem 4 will hold even if the input multimodal distributions $P,Q$ have different numbers of modes. A relatively simple generalization of the statement to the case with an unequal number of modes between $P,Q$ (with $k\neq r$ modes) is to add $r-k$ zero-probability modes to distribution $P$ to match the number of modes between $P,Q$. The zero-probability modes do not affect the theorem’s statement and therefore address a case with an unequal number of modes.
>
> **5- Computational complexity of proposed scores vs. baselines**
>
> **Re:** For the RKE score, we follow the randomized algorithm of estimating the expected value in Theorem 5 that leads to $O(nd)$ computational complexity for $n$ samples of dimension $d$. For the RRKE score, we compute the nuclear norm of the cross-kernel matrix $K_{X,Y}$ through the SVD algorithm which requires $O(\min(mn^2,m^2n)d)$ under $n$ samples for $X$ and $m$ samples for $Y$.
>
> Regarding the baselines, the KID score requires computing the Frobenius norm of the difference of kernel matrices that would cost $O(\max (m^2,n^2) d)$. The FID score requires computing the Frobenius norm of the square root of the empirical covariance matrices which will be $O(\max(n,m)d^3)$. Finally, the precision and recall scores run a k-nearest neighbor search for the empirical samples that need $O(nm\log(m)kd)$ and $O(nmd\log(n)kd)$ computational complexity. We note that a proper computational complexity comparison should take into account how many samples $m,n$ are needed to properly estimate the score for which our results suggest the Gaussian RKE and RRKE can converge faster than the baselines.

---

> > ### Comment · Reviewer_uN6a · 2023-08-17
> >
> > I would like to thank authors for their reply. I also changed my score accordingly. I agree that some new interpretations to the matrix-based Renyi's entropy and the new application is novel. However, I still suggest authors to revise carefully some of their descriptions, and clearly discuss the connections with respect to [Sanchez Giraldo et al., 2014] and [Dong et al., 2023]. After all, the definition or empirical estimator is not novel and has been shown before. Additionally, I also expect authors to put some discussions on methodological differences with respect to [Dong et al., 2023], regarding the part of fast computation.

---

> > > ### Author Response · Authors · 2023-08-20
> > > **Thank you for your feedback**
> > >
> > > We thank Reviewer uN6a for the positive feedback on our response. As discussed in the rebuttal, in the revised draft we will explain the connections between our work and the contributions of (Sanchez Giraldo et al, 2014) and (Dong et al, 2023). We thank the reviewer for pointing out these references in his/her review.

---

### Author Rebuttal · Authors · 2023-08-10

We thank the reviewers for their time and feedback. We have responded to the reviewers' comments and questions under every review textbox. Here, we attach the rebuttal PDF containing the tables and figures discussed in our responses.

---

### Decision · Program_Chairs · 2023-09-21

**Decision:**

Accept (poster)

**Comment:**

The paper presents a novel approach to evaluate the diversity of generative models by using  Rényi entropy. The paper analyses the problem theoretically, show how it measures the number of components of a Gaussian mixture model, as well as experiment on real image data.

The task of evaluating generative models is a very important task, and I believe this paper adds a new insightful perspective on the problem and would be of interest to the community.